# Private Online Learning against an Adaptive Adversary: Realizable and Agnostic Settings

**Bo Li**[1,2]    **Wei Wang**[2]    **Peng Ye**[2]
[1]Guangzhou HKUST Fok Ying Tung Research Institute
[2]Department of Computer Science and Engineering, HKUST
bli@ust.hk, weiwa@cse.ust.hk, pyeac@connect.ust.hk

## Abstract

We revisit the problem of private online learning, in which a learner receives a sequence of $T$ data points and has to respond at each time-step a hypothesis. It is required that the entire stream of output hypotheses should satisfy differential privacy. Prior work of Golowich and Livni [2021] established that every concept class $\mathcal{H}$ with finite Littlestone dimension $d$ is privately online learnable in the realizable setting. In particular, they proposed an algorithm that achieves an $O_d(\log T)$ mistake bound against an oblivious adversary. However, their approach yields a suboptimal $\tilde{O}_d(\sqrt{T})$ bound against an adaptive adversary. In this work, we present a new algorithm with a mistake bound of $O_d(\log T)$ against an adaptive adversary, closing this gap. We further investigate the problem in the agnostic setting, which is more general than the realizable setting as it does not impose any assumptions on the data. We give an algorithm that obtains a sublinear regret of $\tilde{O}_d(\sqrt{T})$ for generic Littlestone classes, demonstrating that they are also privately online learnable in the agnostic setting.

## 1   Introduction

Machine learning has demonstrated remarkable performance in various applications due to its capability of extracting informative patterns from vast amounts of data. However, this success also raises critical privacy concerns, particularly in domains like healthcare or finance, where models often process sensitive personal data. As machine learning technologies continue to advance, ensuring the protection of individual privacy has become an urgent societal and technical challenge.

Differential privacy (DP) [Dwork et al., 2006b,a] is the de facto privacy-preserving technique that addresses these concerns by rigorously formalized privacy guarantees. To ensure that an algorithm protects privacy, DP requires that its output distribution remains nearly indistinguishable when any single individual's data is modified, thereby limiting privacy leakage. The central challenge in differentially private learning lies in designing algorithms that satisfy the DP requirement while remaining effective.

To understand the statistical cost of DP in learning, extensive research has studied probably approximately correct (PAC) learning under DP. A line of works [Alon et al., 2019, Bun et al., 2020, Alon et al., 2022] has established that private learnability is characterized by the Littlestone dimension, a combinatorial measure originally proposed by Littlestone [1988] to describe (non-private) online learnability. In other words, a concept class is privately learnable if and only if it is online learnable.

Motivated by this compelling equivalence, Golowich and Livni [2021] pioneered the study of privately online learning generic concept classes and demonstrated that the equivalence includes private online learnability in the realizable setting. For any concept class $\mathcal{H}$ with Littlestone dimension $d$, their algorithm achieves an $O_d(\log T)$ mistake bound in $T$ rounds against an oblivious adversary that generates

39th Conference on Neural Information Processing Systems (NeurIPS 2025).

the entire data stream prior to interacting with the learner. However, for an adaptive adversary—which dynamically adjusts each data point based on the learner's output history—their approach yields a suboptimal $\tilde{O}_d(\sqrt{T})$ mistake bound. While this upper bound is sufficient to preserve a qualitative equivalence between private and non-private online learning against an adaptive adversary, it leaves open whether adaptive adversaries inherently require higher error rates. Subsequent works [Cohen et al., 2024, Dmitriev et al., 2024, Li et al., 2024] confirmed that a cost of $\Omega(\log T)$ is unavoidable, yet whether $O_d(\log T)$ is achievable for adaptive settings remains unresolved.

Another limitation of their algorithm is that it operates under the realizability assumption, which requires all the data to be perfectly labeled by some $h \in \mathcal{H}$. However, this assumption does not hold in many real-world scenarios, as the labeling function may not belong to $\mathcal{H}$ or even not exist due to noise in data generation. This necessitates the consideration of the *agnostic setting*, where no assumptions are made for the data. Notably, in both (non-private) online learning and private PAC learning, Littlestone classes remain provably learnable in the agnostic setting [Ben-David et al., 2009, Bun et al., 2020, Ghazi et al., 2021b, Beimel et al., 2021, Alon et al., 2020]. This raises a compelling open question: Can this result be generalized to private online learning?

## 1.1 Our Contributions

Our first contribution is an algorithm for private online learning in the realizable adaptive setting with a logarithmic mistake bound.

**Theorem 1.1.** *Let $\mathcal{H}$ be a concept class with Littlestone dimension $d$. In the realizable setting, there exists an $(\varepsilon, \delta)$-differentially private online learner for $\mathcal{H}$ with an expected mistake bound of $O(2^{2^{O(d)}}(\log T + \log(1/\delta))/\varepsilon)$ against any adaptive adversary.*

This result improves upon the previous $\tilde{O}_d(\sqrt{T})$ upper bound established by Golowich and Livni [2021] and addresses an open question they posed. As noted, the logarithmic dependence on $T$ is optimal. However, same as their algorithm, our approach exhibits a doubly exponential dependence on $d$, which is significantly worse than the non-private case [Ghazi et al., 2021b].

We next turn to the agnostic setting. For general Littlestone classes, we show that it is possible to achieve an $\tilde{O}_d(\sqrt{T})$ regret, which is comparable to the non-private case in terms of $T$.

**Theorem 1.2.** *Let $\mathcal{H}$ be a concept class with Littlestone dimension $d$. Then there exists an $(\varepsilon, \delta)$-differentially private online learner for $\mathcal{H}$ with an expected regret of $\tilde{O}(d\sqrt{T}/\varepsilon) + \tilde{O}_d(T^{1/3}/\varepsilon^{2/3})$ against any adaptive adversary in the agnostic setting. When the adversary is oblivious, the regret can be further reduced to $O(\sqrt{dT \log T}) + \tilde{O}_d(T^{1/3}/\varepsilon^{2/3})$.*

As previously discussed, the results of Golowich and Livni [2021] can be interpreted as an equivalence between non-private and private online learning in the realizable setting. The above conclusion generalizes this equivalence to the agnostic setting. Moreover, for an oblivious adversary, the resulting regret matches the best known non-private constructive algorithm [Hanneke et al., 2021] when $\varepsilon \geq \tilde{\Omega}_d(1/T^{1/4})$. Such a "privacy is free" phenomenon has been widely observed by previous works on private OPE (e.g., [Asi et al., 2023b, 2024]). Our result can be viewed as extending this to the nonparametric setting where the class can be infinite (but has finite Littlestone dimension).

## 1.2 Related Work

The investigation of private learning in the PAC framework [Valiant, 1984] was pioneered by Kasiviswanathan et al. [2011]. Following this, a series of studies aimed to characterize the learnability and sample complexity of learning generic concept classes under DP [Beimel et al., 2010, 2019, Feldman and Xiao, 2014, Beimel et al., 2016, Alon et al., 2019, Ghazi et al., 2021b, Alon et al., 2022]. Beimel et al. [2019] demonstrated that, under pure DP, the sample complexity is tightly determined by a measure called the representation dimension. For approximate DP, it was found that the learnability is characterized by the Littlestone dimension [Alon et al., 2019, Bun et al., 2020, Alon et al., 2022]. However, a substantial gap persists between the upper and lower bounds concerning sample complexity [Alon et al., 2019, Ghazi et al., 2021b].

Golowich and Livni [2021]'s work extended private learning to the online model. Building upon the method of Bun et al. [2020] for private PAC learning, they proposed algorithms that attain mistake

bounds sublinear in the time horizon $T$. For the lower bound, several works [Cohen et al., 2024, Dmitriev et al., 2024, Li et al., 2024] discovered that $\Omega(\log T)$ mistakes are necessary under DP. This finding highlights a notable discrepancy between private and non-private settings, as the mistake bound does not grow with $T$ without privacy [Littlestone, 1988]. Whether a stronger separation holds was questioned by Sanyal and Ramponi [2022].

The problem of privately online learning generic concept classes is also closely related to private online prediction from experts (OPE), which has been extensively studied in the literature [Dwork et al., 2010a, Smith and Thakurta, 2013, Jain and Thakurta, 2014, Agarwal and Singh, 2017, Asi et al., 2023a,b, 2024]. While DP-OPE algorithms can be directly applied to finite concept classes, they are not suitable for infinite concept classes with finite Littlestone dimension, which are the focus of this article. Another related problem is private online prediction studied by Kaplan et al. [2023], where the learner only releases a single bit representing the prediction result for the current data point. Under this weaker model, they achieved a better mistake bound compared to the results in [Golowich and Livni, 2021] (in the stronger online learning model) in terms of the Littlestone dimension.

## 2 Preliminaries

We provide some background on online learning, differential privacy, and sanitization in this section.

### 2.1 Online Learning

Online learning can be modeled as a sequential game played between a learner and an adversary. Let $\mathcal{H} \subseteq \{0,1\}^{\mathcal{X}}$ be a concept class over some domain $\mathcal{X}$ and $T$ be an integer indicating the total number of rounds, both of which are known to the learner and the adversary. At each round $t \in [T]$, the learner outputs some hypothesis $h_t : \mathcal{X} \to \{0,1\}$ while at the same time the adversary selects an example $z_t = (x_t, y_t) \in \mathcal{X} \times \{0,1\}$ and presents it to the learner. The performance of the learner is measured by the *regret*, which is the difference between the number of mistakes made by the learner and by the optimal concept in $\mathcal{H}$ (in hindsight), defined as

$$\sum_{t=1}^{T} \mathbb{I}[h_t(x_t) \neq y_t] - \min_{h^{\star} \in \mathcal{H}} \sum_{t=1}^{T} \mathbb{I}[h^{\star}(x_t) \neq y_t].$$

The above scenario is referred to as the *agnostic* setting, where there are no restrictions on the data generated by the adversary. This is in contrast to the *realizable* setting, where there is some $h^{\star} \in \mathcal{H}$ such that $y_t = h^{\star}(x_t)$ for every $t \in [T]$. In this case, the regret is also called the *mistake bound*, as it simply counts the number of mistakes made by the learner. A learner is *proper* if it always outputs $h_t \in \mathcal{H}$ for every $t \in [T]$. Otherwise we say the learner is *improper*.

We consider two variants of adversaries according to their ability of choosing examples: *oblivious* and *adaptive* adversaries. An oblivious adversary can only determine the entire data sequence before interacting with the learner. That is, the data are independent of the learner's internal randomness. In contrast, an adaptive adversary can decide $(x_t, y_t)$ after observing the learner's output history $(h_1, \ldots, h_{t-1})$. Note that in the realizable setting, the adversary does not have to fix in advance an $h^{\star}$ that labels all the data but just needs to ensure the set $\{(x_1, y_1), \ldots, (x_T, y_T)\}$ is consistent with some $h^{\star} \in \mathcal{H}$ at the end of the game. Clearly, an adaptive adversary is more powerful and makes it harder to design an effective learning algorithm.

A learner is considered effective if it always attains a sublinear (i.e., $o(T)$) expected regret. We say a concept class $\mathcal{H}$ is online learnable if there exists such a learner for $\mathcal{H}$. Without privacy, online learnability is characterized by the Littlestone dimension [Littlestone, 1988, Ben-David et al., 2009].

**Definition 2.1** (Shattered Tree). An $\mathcal{X}$-valued tree of depth $n$ is a complete binary $\mathcal{T}$ of depth $n$ (i.e, the number of vertices on any root-to-leaf path is $n$) whose vertices are labeled by elements from $\mathcal{X}$. Every vertex located at the $t$-th layer of $\mathcal{T}$ can be identified by a binary sequence $(y_1, \ldots, y_{t-1}) \in \{0,1\}^{t-1}$ such that it can be reached by starting from the root, then moving to the left child if $y_i = 0$ and to right child if otherwise $y_i = 1$ at step $i \in [t-1]$. For every $t \in [n]$, define $\mathcal{T}_t : \{0,1\}^{t-1} \to \mathcal{X}$ be the mapping from every sequence $(y_1, \ldots, y_{t-1}) \in \{0,1\}^{t-1}$ to the label of the vertex it identifies. We say $\mathcal{T}$ is shattered by $\mathcal{H}$ if for every $(y_1, \ldots, y_n) \in \{0,1\}^n$, there exists $h \in \mathcal{H}$ such that

$$\forall t \in [n], \; h(\mathcal{T}_t(y_1, \ldots, y_{t-1})) = y_t.$$

**Definition 2.2** (Littlestone Dimension). The Littlestone dimension of a concept class $\mathcal{H}$ over $\mathcal{X}$, denoted by $\mathrm{Ldim}(\mathcal{H})$, is the largest $d$ such that there is $\mathcal{X}$-valued tree $\mathcal{T}$ of depth $d$ shattered by $\mathcal{H}$.

One can also view $\mathcal{X}$ as a concept class over domain $\mathcal{H}$ by defining $x(h) = h(x)$ for any $x \in \mathcal{X}$ and $h \in \mathcal{H}$. This class $\mathcal{X}$ is called the dual class of $\mathcal{H}$. The dual Littlestone dimension of $\mathcal{H}$, denoted by $\mathrm{Ldim}^\star(\mathcal{H})$, is defined as the Littlestone dimension of the dual class $\mathcal{X}$.

In the realizable setting, it was shown by Littlestone [1988] that the best attainable mistake bound is exactly $\mathrm{Ldim}(\mathcal{H})$ for deterministic learners.[1] The mistake bound is achieved by an algorithm called the *Standard Optimal Algorithm* (SOA) that makes at most $\mathrm{Ldim}(\mathcal{H})$ mistakes on any realizable sequence. Like the work of Golowich and Livni [2021], we will access the SOA as a black box and our algorithm only relies on the fact that the SOA has a mistake bound of $\mathrm{Ldim}(\mathcal{H})$.

We next introduce the online prediction from experts (OPE) problem. In this problem, there are $N$ experts. At each round $t$, the algorithm chooses an expert $i_t \in [N]$ while the adversary chooses a loss function $\ell_t : [N] \to [0,1]$. Then the function $\ell_t$ is released to the algorithm and a loss of $\ell_t(i_t)$ is incurred. The regret of the algorithm is defined as

$$\sum_{t=1}^{T} \ell_t(i_t) - \min_{i \in [N]} \sum_{t=1}^{T} \ell_t(i).$$

Similar to online learning, an oblivious adversary can only choose $(\ell_1, \ldots, \ell_T)$ at the very beginning while an adaptive adversary can choose $\ell_t$ after seeing $(i_1, \ldots, i_{t-1})$.

## 2.2 Differential Privacy

We start by recalling the classical notion of differential privacy. Let $\mathcal{Z}$ be some data domain ($\mathcal{Z} = \mathcal{X} \times \{0,1\}$ in online learning). Let $S = (z_1, \ldots, z_T) \in \mathcal{Z}^T$ and $S' = (z'_1, \ldots, z'_T) \in \mathcal{Z}^T$ be two data sequences of length $T$. We say $S$ and $S'$ are neighboring if they differ in at most one entry, i.e., there exists some $i$ such that $z_j = z'_j$ for all $j \in [T] \setminus \{i\}$.

**Definition 2.3** (Differential Privacy). A randomized algorithm $\mathcal{A}$ is $(\varepsilon, \delta)$-differentially private if for any pair of neighboring data sequences $S, S' \in \mathcal{Z}^T$ and any set $O$ of outputs, we have

$$\Pr[\mathcal{A}(S) \in O] \le e^\varepsilon \Pr[\mathcal{A}(S') \in O] + \delta.$$

The above standard definition of differential privacy cannot capture the scenario that the data sequence is adaptively generated by an adversary. We next rigorously define differential privacy in the presence of adaptive inputs following the formulation of Jain et al. [2023]. Consider a $T$-round game played between an algorithm $\mathcal{A}$ and an adversary $\mathcal{B}$, where $\mathcal{A}$ presents some $h_t$ to $\mathcal{B}$ and receives some data $z_t$ from $\mathcal{B}$ at every round $t$. The adversary $\mathcal{B}$ can (adaptively) choose one special round $t^\star \in [T]$. At this round, $\mathcal{B}$ generates two data points $z_{t^\star}^{(0)}$ and $z_{t^\star}^{(1)}$. Then $z_{t^\star}^{(b)}$ will be sent to $\mathcal{A}$, where $b \in \{0,1\}$ is some global parameter that is *unknown* to both $\mathcal{A}$ and $\mathcal{B}$. Let $\Pi_{\mathcal{A},\mathcal{B}}(b)$ denote $\mathcal{B}$'s view of the game, including $(h_1, \ldots, h_T)$ and the internal randomness of $\mathcal{B}$. To ensure privacy, we require that $\mathcal{B}$ is unlikely to tell the value of $b$, formalized as follows.

**Definition 2.4** (Differential Privacy with Adaptive Inputs). A randomized algorithm $\mathcal{A}$ is $(\varepsilon, \delta)$-differentially private if for any adversary $\mathcal{B}$ and any set $O$ of views, we have

$$\Pr[\Pi_{\mathcal{A},\mathcal{B}}(0) \in O] \le e^\varepsilon \Pr[\Pi_{\mathcal{A},\mathcal{B}}(1) \in O] + \delta.$$

Following the common treatment of privacy parameters in private learning Dwork et al. [2014], Bun et al. [2020], we will assume throughout this article that $\varepsilon$ is at most some small constant (say, $0.1$) and $\delta$ is significantly smaller than the reciprocal of the time horizon (i.e., $\delta = T^{-\omega(1)}$). We say a concept class $\mathcal{H}$ is privately online learnable in the realizable (or agnostic) setting if there is an $(\varepsilon, \delta)$-differentially private algorithm that attains a sublinear *expected* mistake bound (or regret) with $\varepsilon \le 0.1$ and $\delta = T^{-\omega(1)}$.

We next present some useful tools to achieve differential privacy. The first is the AboveThreshold mechanism [Dwork et al., 2009]. Given a sequence of sensitivity-1 data point, the AboveThreshold mechanism allows us to privately monitor whether the cumulative sum exceeds some threshold.

---

[1]For randomized learners, the optimal expected mistake bound is equal to the randomized Littlestone dimension of $\mathcal{H}$ [Filmus et al., 2023], which is between $\mathrm{Ldim}(\mathcal{H})/2$ and $\mathrm{Ldim}(\mathcal{H})$.

**Theorem 2.5** ([Dwork et al., 2009, 2014]). *Let $T$ be the time horizon, $\varepsilon$ be the privacy parameter, and $\tau$ be some threshold value. There exists an $(\varepsilon, 0)$-differentially private algorithm AboveThreshold that at each round $t \in [T]$ receives some $b_t \in [0, 1]$ (may be chosen adaptively) and responds an $a_t \in \{\top, \bot\}$ such that with probability $1 - \beta$, we have:*

- *For all $a_t = \top$, it holds that $\sum_{i=1}^{t} b_i \geq \tau - \frac{8(\ln T + \ln(2/\beta))}{\varepsilon}$.*

- *For all $a_t = \bot$, it holds that $\sum_{i=1}^{t} b_i \leq \tau + \frac{8(\ln T + \ln(2/\beta))}{\varepsilon}$.*

For a dataset $S = (z_1, \ldots, z_n)$, let $\mathrm{Count}_S(z)$ denote the number of occurrence of $z$ in $S$, i.e., $\mathrm{Count}_S(z) = \sum_{i \in [n]} \mathbb{I}[z_i = z]$. We can use the PrivateHistogram algorithm to privately publish $\mathrm{Count}_S$. The problem was studied in the context of sanitization in [Beimel et al., 2016, Bun et al., 2019]. Here we adopt the algorithm from [Aliakbarpour et al., 2024] as the resulting error bound is easier to work with.

**Theorem 2.6** (Private Histogram [Aliakbarpour et al., 2024]). *Let $S$ be a dataset over $\mathcal{Z}$. There exists an $(\varepsilon, \delta)$-differentially private algorithm that outputs a function $\overline{\mathrm{Count}}_S : \mathcal{Z} \to \mathbb{R}$ such that with probability $1$ we have*

$$\sup_{z \in \mathcal{Z}} \left| \overline{\mathrm{Count}}_S(z) - \mathrm{Count}_S(z) \right| \leq \frac{8 \ln(8/\delta)}{\varepsilon}.$$

### 2.3 Sanitization

Let $S = (x_1, \ldots, x_n) \in \mathcal{X}^n$ be a dataset. For any $h \in \mathcal{H}$, define $\hat{P}_S(h) = \frac{1}{n} \sum_{i=1}^{n} h(x_i)$. The task of sanitization is to estimate $\hat{P}_S(h)$ for every $h \in \mathcal{H}$.

**Definition 2.7** ([Blum et al., 2013, Beimel et al., 2016]). *Let $\mathcal{H}$ be a concept class over $\mathcal{X}$. An $(\alpha, \beta)$-sanitizer for $\mathcal{H}$ takes as input a dataset $S \in \mathcal{X}^n$ and outputs a function $\mathrm{Est} : \mathcal{H} \to [0, 1]$ such that with probability $1 - \beta$ it holds that $\sup_{h \in \mathcal{H}} |\mathrm{Est}(h) - \hat{P}_S(h)| \leq \alpha$.*

Note that in the above definition we only require the sanitizer to output a function rather than a sanitized dataset. But one can always use it to generate a synthetic dataset by finding an $S'$ such that $|\mathrm{Est}(h) - \hat{P}_{S'}(h)| \leq \alpha$ for all $h \in \mathcal{H}$. With probability $1 - \beta$, such an $S'$ is guaranteed to exist since the input $S$ satisfies this property. By the triangle inequality, we have $|\hat{P}_{S'}(h) - \hat{P}_S(h)| \leq 2\alpha$. Therefore, we can also assume a sanitizer directly outputs a sanitized dataset with error $2\alpha$.

Sometimes we may want to sanitize a labeled dataset $S \in (\mathcal{X} \times \{0, 1\})^n$ with respect to an extended class $\mathcal{H}^{\mathrm{label}} = \{h^{\mathrm{label}} : h \in \mathcal{H}\}$ over $\mathcal{X} \times \{0, 1\}$, where $h^{\mathrm{label}} : \mathcal{X} \times \{0, 1\} \to \{0, 1\}$ is the predicate indicating whether $h$ makes an error, i.e., $h^{\mathrm{label}}((x, y)) = \mathbb{I}[h(x) \neq y]$. The following lemma demonstrates that a sanitizer for $\mathcal{H}$ can be converted to one for $\mathcal{H}^{\mathrm{label}}$.

**Lemma 2.8** ([Bousquet et al., 2020]). *Suppose there is an $(\varepsilon, \delta)$-differentially private $(\alpha, \beta)$-sanitizer for $\mathcal{H}$ with input size $n$. Then there exists an $(O(\varepsilon), O(\delta))$-differentially private $(O(\alpha), O(\beta))$-sanitizer for $\mathcal{H}^{\mathrm{label}}$ with input size $n$ as long as $n \geq C \ln(1/\beta)/\varepsilon\alpha$ for some constant $C$.*

## 3 Realizable Online Learning

In this section, we present our realizable learner that achieves a logarithmic mistake bound against an adaptive adversary. For clarity, we denote by $\mathcal{H}$ the given concept class and by $d$ its Littlestone dimension. For a sequence $S$ of length $t$, we write $\mathrm{SOA}(S)$ to represent the hypothesis that the SOA will output at time-step $t + 1$.

We start by reiterating the method of Golowich and Livni [2021] and analyzing why it fails to provide a logarithmic mistake bound in the presence of an adaptive adversary. Their algorithm creates a forest consisting of sufficiently many binary trees of depth $d$ and maintains a set of nodes called pertinent nodes. Initially, every leaf node is pertinent and is associated with an empty sequence. At each round, the learner randomly selects a pertinent node and inserts the input example into the sequence assigned to this node. After that, an update step is performed.

The update procedure follows the key idea of constructing tournament examples in [Bun et al., 2020]. Let $S_1$ and $S_2$ be two sequences associated with two pertinent sibling nodes. Once it becomes the

case that $\text{SOA}(S_1) \neq \text{SOA}(S_2)$, the algorithm chooses some $\bar{x}$ such that $\text{SOA}(S_1)(\bar{x}) \neq \text{SOA}(S_2)(\bar{x})$ and guesses its label $\bar{y} \in \{0, 1\}$ randomly. Suppose the SOA predicts the label of $\bar{x}$ incorrectly as $1 - \bar{y}$ on $S_k$ ($k \in \{0, 1\}$). Then a new sequence is created by appending the pair $(\bar{x}, \bar{y})$ to $S_k$. The two sibling nodes are removed from the set of pertinent nodes while their parent becomes pertinent and is associated with the new sequence. The algorithm then recursively performs the update on their parent until reaching a node whose sibling is not pertinent.

Suppose the input sequence is fixed and let $h^\star \in \mathcal{H}$ be the labeling function. They observed that the random insertion of the examples is equivalent to a random permutation on every layer of the forest. Based on this observation, they proved that among the hypotheses produced by running the SOA on the sequences assigned to pertinent nodes, with high probability there exists at least a frequent one. They designed a mechanism that privately releases a frequent hypothesis at each round with logarithmic cost.

Since the output hypothesis is frequent at every round, once the algorithm makes a mistake, with some positive probability the state of the SOA on some sequence will change and an update will be performed. Note that $h^\star(\bar{x}) = \bar{y}$ with probability $1/2$. Therefore, for every tree the SOA will output $h^\star$ at the root with probability roughly $1/2^{2^d}$. As long as the number of trees is sufficiently large, the algorithm is able to identify $h^\star$ privately. As a result, the total number of mistakes can be bounded by the number of nodes in the forest.

In the presence of an adaptive adversary, there are two main obstacles in applying their algorithm:

- The output at each round partially reveals the information about the random assignment of examples. This disqualifies their random permutation argument in proving the existence of frequent hypotheses as it requires the examples and the random insertion to be independent.
- The labeling function $h^\star \in \mathcal{H}$ is not fixed in advance. Then one cannot simply conclude that every tournament example is correct with probability $1/2$.

In their work, they resort to a standard reduction [Cesa-Bianchi and Lugosi, 2006] that transforms a learner against an oblivious adversary to one against an adaptive adversary. However, the reduction requires running a new instance from the beginning at each round, incurring a mistake bound of $\sqrt{T}$ due to advanced composition [Dwork et al., 2010b] of approximate DP.

We next illustrate how we tackle these two challenges to obtain a logarithmic regret. We address the first one by a *lazy update* technique and the second one by the *uniform convergence* argument.

**Lazy update.** Unlike their algorithm, which performs the update immediately, we delay the update until there are enough collisions (i.e., sibling nodes with sequences on which the SOA outputs differently) in one layer. Once the condition is met, we update the whole layer and proceed to the upper layer. We then perform a random permutation in order to leverage their argument. Since the randomness is independent of the examples in this process, their argument can be successfully applied.

**Uniform convergence.** Since the labeling function $h^\star$ is not predetermined, we have to argue that the number of trees consistent with $h$ is sufficiently large simultaneously for every $h \in \mathcal{H}$ that is consistent with the data we have seen so far. However, one cannot directly apply the union bound since there can be infinitely many feasible labeling functions. To circumvent this, we observe that the number of data points in the forest (including input data points and tournament examples we generate) is bounded. We can then prove the result by a classical uniform convergence argument [Vapnik and Chervonenkis, 1971] over $\mathcal{H}$.

We present our update subroutine in Algorithm 1. We use the symbol $\perp$ for the case that the SOA fails on a non-realizable sequence. In the $s$-th layer, there are $N_s$ sequences $S_1^s, \ldots, S_{N_s}^s$, where $S_{2i-1}^s$ and $S_{2i}^s$ ($i \in [N_s/2]$) are considered as sibling sequences. In the update procedure, we will create a new sequence from every pair of sibling sequences by padding a tournament example. The new sequence is then placed in a random location in the next layer, i.e., $S_{\pi(i)}^{s+1}$.

We next describe how the entire algorithm works. In the $s$-th layer, the algorithm maintains $N_s$ sequences and a list $L_s$ of frequent hypotheses. We keep outputting a hypothesis from $L_s$ and run an instance of AboveThreshold to inspect the number of mistakes. Once the number exceeds a particular threshold, we switch to the next frequent hypothesis with a new instance of AboveThreshold. We also insert the data point received at each round into a random sequence. After iterating over all

---

**Algorithm 1:** Update

---

**Global Parameter:** concept class $\mathcal{H}$
**Input:** sequences $S_1^s, \ldots, S_{N_s}^s$

1   $N_{s+1} \leftarrow N_s/2$.
2   Create $S_1^{s+1}, \ldots, S_{N_{s+1}}^{s+1}$ such that every $S_i^{s+1}$ is initialized as $\perp$.
3   Let $\pi$ be a random permutation over $[N_{s+1}]$.
4   **for** $i = 1, \ldots, N_{s+1}$ **do**
5      **if** $S_{2i-1}^s \neq \perp$ *and* $S_{2i}^s \neq \perp$ *and* $\mathrm{SOA}(S_{2i-1}^s) \neq \mathrm{SOA}(S_{2i}^s)$ **then**
6          Pick $\bar{x}_i$ such that $\mathrm{SOA}(S_{2i-1}^s)(\bar{x}_i) \neq \mathrm{SOA}(S_{2i}^s)(\bar{x}_i)$ and draw $\bar{y}_i$ from $\{0, 1\}$ uniformly.
7          $S_{\pi(i)}^{s+1} \leftarrow (S_j^s, (\bar{x}_i, \bar{y}_i))$ where $j \in \{2i-1, 2i\}$ is such that $\mathrm{SOA}(S_j^s)(\bar{x}_i) \neq \bar{y}_i$.
8      **end**
9   **end**
10   Output $S_1^{s+1}, \ldots, S_{N_{s+1}}^{s+1}$

---

frequent hypotheses in $L_s$, we perform an update, invoke PrivateHistogram to filter all new frequent hypotheses out, and repeat the same procedure for $L_{s+1}$. The details are presented in Algorithm 2.

---

**Algorithm 2:** Realizable learner

---

**Global Parameter:** time horizon $T$, concept class $\mathcal{H}$, privacy parameters $\varepsilon, \delta$, failure
                   probability $\beta$, initial number of nodes $N_0$
**Input:** input sequence $((x_1, y_1), \ldots, (x_T, y_T))$

1   $d \leftarrow \mathrm{Ldim}(\mathcal{H})$, $s \leftarrow 0$, $\varepsilon_0 \leftarrow \varepsilon/2$, $N_i \leftarrow N_0/2^i$ for $i \in [d]$.
2   Create $S_1^0, \ldots, S_{N_0}^0$ such that every $S_i^0$ is initialized as $\emptyset$.
3   Create a list $L_0 \leftarrow \{\mathrm{SOA}(\emptyset)\}$.
4   Initiate an instance of AboveThreshold with privacy parameter $\varepsilon_0$ and threshold
     $N_0 + \frac{8(\ln T + \ln(6T/\beta))}{\varepsilon_0}$.
5   **for** $t = 1, \ldots, T$ **do**
6      Set $h_t$ to be the first element in $L_s$ and output $h_t$, halt if $L_s$ is empty.
7      Sample $i_t$ uniformly from $[N_s/2]$.
8      **if** $\mathrm{SOA}(S_{2i_t-1}^s) = \mathrm{SOA}(S_{2i_t}^s) \neq \perp$ *and* $\mathrm{SOA}(S_{2i_t-1}^s)(x_t) \neq y_t$ **then**
9          $S_{2i_t-1}^s \leftarrow (S_{2i_t-1}^s, (x_t, y_t))$.
10      **end**
11      Feed $\mathbb{I}[h_t(x_t) \neq y_t]$ to AboveThreshold and receive $a_t \in \{\top, \perp\}$.
12      **if** $a_t = \top$ **then**
13          Halt the current AboveThreshold and remove the first element in $L_s$.
14          **while** $s < d$ *and* $L_s$ *is empty* **do**
15              Feed $S_1^s, \ldots, S_{N_s}^s$ to Update and receive $S_1^{s+1}, \ldots, S_{N_{s+1}}^{s+1}$.
16              $s \leftarrow s + 1$.
17              Create a multiset
                 $V_s \leftarrow \{\mathrm{SOA}(S_{2i}^s) : i \in [N_s/2] \text{ and } \mathrm{SOA}(S_{2i-1}^s) = \mathrm{SOA}(S_{2i}^s) \neq \perp\}$.
18              Run PrivateHistogram with privacy parameters $(\varepsilon_0/d, \delta/d)$ on $V_s$ and obtain
             $\overline{\mathrm{Count}}_{V_s}$.
19              Set $L_s \leftarrow \{h : \overline{\mathrm{Count}}_{V_s}(h) \geq 3M_s/4\}$, where $M_s = 128 \cdot 2^{-6 \cdot 2^s} N_s$.
20          **end**
21          Initiate a new instance of AboveThreshold with privacy parameter $\varepsilon_0$ and threshold
         $N_s + \frac{8(\ln T + \ln(6T/\beta))}{\varepsilon_0}$.
22      **end**
23   **end**

---

It is not hard to see that the algorithm preserves privacy. For utility, note that PrivateHistogram extracts all frequent hypotheses in layer $s$ and stores them in $L_s$. Suppose $h \in L_s$ can be obtained

by running the SOA on $M_s$ pairs of sibling sequences. By the property of AboveThreshold, we will output $h$ until it makes roughly $N_s$ mistakes. Since every data point is inserted uniformly at random, classical results of the coupon collector's problem ensure that at least $M_s/2$ pairs are covered and become collisions (i.e., $\texttt{SOA}(S_{2i-1}^s) \neq \texttt{SOA}(S_{2i}^s)$). Once $L_s$ is exhausted, we proceed to the next layer by invoking the update subroutine. By leveraging the idea of Golowich and Livni [2021] and the uniform convergence argument, we can prove that for every $h \in \mathcal{H}$ that is consistent with the data we have seen so far, after the update there are $M_{s+1} = CM_s^2/N_{s+1}$ ($C$ is some constant) pairs of sibling sequences that are still consistent with $h$ and, either they are already collisions or running the SOA on them gives the same hypothesis $h_0$. This allows us to act recursively until $s = d$, which indicates $L_d$ contains all the possible labeling functions. Solving the recurrence relation gives $N_0 \approx 2^{O(2^d)}$, which yields the desired mistake bound.

We formally state our results in the following theorem. A detailed proof is given in Appendix B. Setting the failure probability $\beta = 1/T$ directly yields Theorem 1.1.

**Theorem 3.1.** *Let $\mathcal{H}$ be a concept class with Littlestone dimension $d$. Algorithm 2 with parameter $N_0 = 2^{\Theta(2^d)}(\ln(T/\beta) + \ln(1/\delta)/\varepsilon)$ is an $(\varepsilon, \delta)$-differentially private online learner that makes at most*

$$O\left(\frac{2^{O(2^d)}(\log T + \log(1/\beta) + \log(1/\delta))}{\varepsilon}\right)$$

*mistakes with probability $1 - \beta$.*

We remark that the SOA can be replaced by any deterministic online learner with bounded number of mistakes. For classes that can be properly learned by a deterministic online learner (e.g., thresholds over finite domain), our algorithm can be made proper as well. However, there are simple examples suggesting that randomness is necessary for proper online learning (see, e.g., [Hanneke et al., 2021]). Hence, our algorithm is improper in general.

# 4   Agnostic Online Learning

In this section, we present our algorithms in the agnostic setting. We first give a simple proper learner with a suboptimal regret. Then we show how to improve the regret to $\tilde{O}_d(\sqrt{T})$ (but result in an improper learner).

## 4.1   A Simple Algorithm Using Sanitization

In our algorithm, we divide the entire sequence into batches of size $B$. After each batch, we invoke a sanitizer for $\mathcal{H}^{\text{label}}$ to obtain synthetic examples. All the synthetic data are partitioned into $B$ disjoint subsequences, where each subsequence contains exactly one data point from each batch. Our prediction at each round is determined by running a (non-private) online learner on one of the disjoint subsequences. The overall framework is depicted in Algorithm 3. Similar to the SOA, we write $\mathcal{A}(S)$ to denote the output distribution of $\mathcal{A}$ after inputting a sequence $S$.

**Theorem 4.1.** *Let $\mathcal{A}$ be a (non-private) proper online learner with expected regret $R(T')$ for any time horizon $T'$ and $\mathcal{B}$ be an $(\varepsilon, \delta)$-differentially private $(\alpha, \beta)$-sanitizer for $\mathcal{H}^{\text{label}}$ with input size $B$. Then Algorithm 3 is an $(\varepsilon, \delta)$-differentially private proper online learner that attains an expected regret of $O(B \cdot R(T/B) + \alpha T)$ against an adaptive adversary conditioned on some event $E$ with $\Pr[E] \geq 1 - T/B \cdot \beta$.*

We now leverage the sanitizer for generic Littlestone classes proposed by Ghazi et al. [2021b]. The original statement exhibits a sample complexity of $\tilde{O}(d^6\sqrt{d^\star}/\varepsilon\alpha^3)$, which was obtained by applying their private proper agnostic learner to the synthetic data generator proposed by Bousquet et al. [2020]. But we can save a factor of $1/\alpha$ through a few tweaks to the algorithm and analysis:

- The sample complexity of the private proper PAC (realizable) learner [Ghazi et al., 2021b] can be improved to $\tilde{O}(d^6/\varepsilon\alpha)$ by replacing the uniform convergence argument in their proof with a weaker relative uniform convergence argument.

- The discriminator of [Bousquet et al., 2020] can be implemented using a private agnostic empirical learner, which does not incur the $\tilde{O}_d(1/\alpha^2)$ generalization cost.

---
**Algorithm 3:** A simple private online learner
---
**Global Parameter:** time horizon $T$, concept class $\mathcal{H}$, batch size $B$
**Input:** online learner $\mathcal{A}$, sanitizer $\mathcal{B}$ for $\mathcal{H}^{\text{label}}$, input sequence $((x_1, y_1), \ldots, (x_T, y_T))$

1   Initialize $S_1^1, \ldots, S_B^1$ as $\emptyset$.
2   **for** $t = 1, \ldots, T$ **do**
3      Let $b = \lceil t/B \rceil$ be the batch index.
4      Draw $i_t$ uniformly from $[B]$, output $h_t$ where $h_t \sim \mathcal{A}(S_{i_t}^b)$.
5      **if** $t \equiv 0 \pmod{B}$ **then**
6          Run $\mathcal{B}$ on $((x_{t-B+1}, y_{t-B+1}), \ldots, (x_t, y_t))$ and construct synthetic data
            $((x'_{t-B+1}, y'_{t-B+1}), \ldots, (x'_t, y'_t))$, exit if fail.
7          Perform a random permutation over $((x'_{t-B+1}, y'_{t-B+1}), \ldots, (x'_t, y'_t))$.
8          $S_i^{b+1} \leftarrow (S_i^b, (x'_{t-B+i}, y'_{t-B+i}))$ for every $i \in [B]$.
9      **end**
10   **end**
---

We provide a detailed discussion in Appendix D and present the final result below.

**Theorem 4.2** ([Bousquet et al., 2020] and [Ghazi et al., 2021b], Strengthened). *Let $\mathcal{H}$ be a concept class with Littlestone dimension $d$ and dual Littlestone dimension $d^\star$. Then there exists an $(\varepsilon, \delta)$-differentially private $(\alpha, \beta)$-sanitizer for $\mathcal{H}$ with sample complexity $\tilde{O}(d^6 \sqrt{d^\star}/\varepsilon \alpha^2)$.*

A combination of Theorem 4.1, Theorem 4.2, Lemma 2.8, and regret bounds for proper online learning [Alon et al., 2021, Hanneke et al., 2021] yields the following regret bound for private online learning. Since $d^\star \leq 2^{2^{d+2}} - 2$ [Bhaskar, 2021], it implies that every Littlestone class is privately (and properly) online learnable in the agnostic setting.

**Corollary 4.3.** *Let $\mathcal{H}$ be a concept class with Littlestone dimension $d$ and dual Littlestone dimension $d^\star$. Then there exists an $(\varepsilon, \delta)$-differentially private proper online learner for $\mathcal{H}$ with an expected regret of $\tilde{O}(T^{3/4} \cdot (d^7 \sqrt{d^\star}/\varepsilon)^{1/4})$ against an adaptive adversary.*

### 4.2   Online Learning via Privately Constructing Experts

We have shown a private online learner with regret $\tilde{O}_d(T^{3/4})$. However, even if we have a sanitizer with error $\alpha = 1/B$, optimizing the choice of $B$ (i.e., $B = \Theta_d(T^{1/3})$) gives an $O_d(\sqrt{TB} + T/B) = O_d(T^{2/3})$ regret, which is still significantly worse than the $O_d(\sqrt{T})$ bound in the non-private case.

To break this barrier, we exploit the approach of constructing experts, which was proposed by Ben-David et al. [2009] for agnostic online learning. The idea is based on the fact that for any $h \in \mathcal{H}$ the SOA makes at most $d$ mistakes on a sequence relabeled by $h$. Hence one can enumerate the rounds at which the SOA makes mistakes and use the SOA to simulate the behavior of $h$ on the entire input sequence. Then an $\tilde{O}(\sqrt{dT})$ regret can be achieved by creating $\binom{T}{\leq d} = O(T^d)$ instances of the SOA as experts and running an OPE algorithm [Littlestone and Warmuth, 1994].

Since the constructed experts heavily depend on the input data, directly employing the same method would violate privacy. Therefore, we again resort to the idea of sanitization. By incorporating the binary mechanism for continual observation [Dwork et al., 2010a], we can detect if a concept makes more than $\tilde{O}_d(\sqrt{T})$ mistakes in a time interval. We can enumerate the $d$ endpoints that decide the intervals. Since we also have to enumerate which sanitized data points are fed to the SOA, the number of experts will grow by some amount, but remains adequate to run a private OPE algorithm.

An issue of the above construction is that the output hypothesis of the SOA may not belong to $\mathcal{H}$ and its mistakes cannot be observed on sanitized data. Moreover, the structure of the output hypotheses of the SOA is hard to characterize. We bypass this by replacing the SOA with the online learner proposed by Hanneke et al. [2021]. Their online learner has a slightly larger mistake bound of $O(d)$, but the output is guaranteed to be the majority of very few concepts in $\mathcal{H}$. Thus, we only have to sanitize a moderately larger class. Now we have a set of experts such that at least one of them is no worse than the optimal $h^\star \in \mathcal{H}$ by $\tilde{O}_d(\sqrt{T})$. This allows us to run any private OPE algorithm to obtain a private online learner for $\mathcal{H}$.

Note that we can further reduce the number of mistakes made by the experts if we have a sanitizer with a better dependence on $\alpha$. Though we don't know if the current sample complexity of sanitization can be improved, we can still refine the above result by an alternative approach. This is due to an observation that we only need to detect if an expert already made a lot of mistakes rather than an accurate estimate of the number of mistakes. We design an algorithm for this problem with sample complexity $\tilde{O}_d(1/\alpha^{1.5})$ based on Bousquet et al. [2020]'s framework, thereby achieving a better regret. We present below a simplified statement of our final result, which implies Theorem 1.2 by applying existing algorithms for private OPE [Jain and Thakurta, 2014, Asi et al., 2024]. The detailed results and proofs can be found in Appendix E.

**Theorem 4.4.** *Let $\mathcal{H}$ be a concept class with Littlestone dimension $d$. Suppose there exists an $(\varepsilon, \delta)$-differentially private algorithm for the OPE problem with $N$ experts and time horizon $T$ that has an expected regret of $R(\varepsilon, \delta, T, N)$. Then there exists a $(2\varepsilon, 2\delta)$-differentially private online learner for $\mathcal{H}$ with an expected regret of $R(\varepsilon, \delta, T, N) + \tilde{O}_d(T^{1/3}/\varepsilon^{2/3})$. Furthermore, if the regret bound for the OPE problem holds against an adaptive adversary, and the resulting regret bound for online learning $\mathcal{H}$ also holds against an adaptive adversary.*

## 5  Discussion and Future Work

In this work, we study online learning under differential privacy. For the realizable setting, we propose an algorithm with an $O_d(\log T)$ mistake bound against any adaptive adversary, which significantly improves the previous result of Golowich and Livni [2021] and achieves an optimal dependence on $T$. For the agnostic setting, our method yields a regret of $\tilde{O}_d(\sqrt{T})$, which achieves nearly the same rate as the non-private case in terms of $T$ up to logarithmic factors.

We discuss some potential future directions below.

**Proper private online learning.** Our optimal algorithms for realizable and agnostic settings are improper. For the realizable setting, it is known that a mistake bound of $O_d(\text{polylog}(T))$ is attainable without privacy [Daskalakis and Golowich, 2022]. We ask if this is also possible under differential privacy. For the agnostic setting, our Algorithm 3 has a suboptimal $\tilde{O}_d(T^{3/4})$ regret. We believe this can be improved to $\tilde{O}_d(\sqrt{T})$ and leave it as future work.

**Dependence on $d$.** All of our algorithms incur a doubly exponential dependence on $d$. We wonder if the dependence can be improved to polynomial as in private PAC learning [Ghazi et al., 2021b].[2]

**Unknown horizon $T$.** In this work, we assume the time horizon $T$ is known in advance. This requirement can be removed by the classical doubling trick — splitting the input sequence into buckets of lengths $1, 2, 4, 8, \cdots$ and running our algorithm on each bucket separately (with $T = 1, 2, 4, 8, \cdots$). This does not affect our regret bound for the agnostic setting but will increase the mistake bound of our realizable learner (Algorithm 2) to $O_d(\log^2 T)$. Whether a mistake bound of $O_d(\log T)$ is still achievable without knowing $T$ in advance is an interesting question.

## Acknowledgments and Disclosure of Funding

The research was supported in part by an NSFC grant 62432008, RGC RIF grant R6021-20, an RGC TRS grant T43-513/23N-2, RGC CRF grants C7004-22G, C1029-22G and C6015-23G, NSFC/RGC grant CRS_HKUST601/24 and RGC GRF grants 16207922, 16207423, 16203824 and 16211123.

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

# A  Additional Preliminaries

## A.1  PAC Learning

Let $P$ be a distribution over domain $\mathcal{X}$ and $h$ be a hypothesis, we write $P(h)$ to denote $\mathbb{E}_{x\sim P}[h(x)]$. For an unlabeled dataset $S \in \mathcal{X}^n$, we write $\hat{P}_S$ to denote the empirical distribution over $S$. Given two hypotheses $h_1$ and $h_2$, we define $h_1 \oplus h_2$ as the hypothesis such that $(h_1 \oplus h_2)(x) = \mathbb{I}[h_1(x) \neq h_2(x)]$ for all $x \in \mathcal{X}$. For two hypothesis classes $\mathcal{H}_1$ and $\mathcal{H}_2$, define $\mathcal{H}_1 \oplus \mathcal{H}_2 = \{h_1 \oplus h_2 : h_1 \in \mathcal{H}_1,\ h_2 \in \mathcal{H}_2\}$. The *generalization disagreement* between $h_1$ and $h_2$ with respect to $P$ is defined as $\mathrm{dis}_P(h_1, h_2) = P(h_1 \oplus h_2)$. The *empirical disagreement* between $h_1$ and $h_2$ is defined as $\mathrm{dis}_S(h_1, h_2) = \mathrm{dis}_{\hat{P}_S}(h_1, h_2)$.

We then take into account the labels. For a distribution $P$ over $\mathcal{X} \times \{0, 1\}$. The *generalization error* of a hypothesis $h$ with respect to $P$ is defined as $\mathrm{err}_P(h) = \Pr_{(x,y)\sim P}[h(x) \neq y]$. Recall that $h^{\mathrm{label}}((x, y)) = \mathbb{I}[h(x) \neq y]$, we have $\mathrm{err}_P(h) = P(h^{\mathrm{label}})$. For a labeled dataset $S \in (\mathcal{X} \times \{0, 1\})^n$, the *empirical error* of $h$ with respect to $S$ is defined as $\mathrm{err}_S(h) = \mathrm{err}_{\hat{P}_S}(h)$.

We now introduce the PAC learning model. In this model, the learner takes as input a labeled dataset $S$ with each element sampled from some unknown distribution $P$. Moreover, it is guaranteed that there exists some $h^\star \in \mathcal{H}$ that labels all the data points. The task of the leaner is to find a hypothesis that minimizes the generalization error.

**Definition A.1** (PAC Learning [Valiant, 1984]). An algorithm $\mathcal{A}$ is said to be an $(\alpha, \beta)$-PAC learner for concept class $\mathcal{H}$ with sample complexity $n$ if for any distribution $P$ over $\mathcal{X} \times \{0, 1\}$ such that $\Pr_{(x,y)\sim P}[h^\star(x) = y] = 1$ for some $h^\star \in \mathcal{H}$, it takes as input a dataset $S = ((x_1, y_1), \ldots, (x_n, y_n))$, where every $(x_i, y_i)$ is drawn i.i.d. from $P$, and outputs a hypothesis $h$ satisfying

$$\Pr[\mathrm{err}_P(h) \leq \alpha] \geq 1 - \beta,$$

where the probability is taken over the random generation of $S$ and the random coins of $\mathcal{A}$.

In contrast to PAC learning, the agnostic learning model [Haussler, 1992, Kearns et al., 1994] imposes no assumptions on the underlying distribution. The objective is to identify a hypothesis whose generalization error is close to that of the best one in $\mathcal{H}$.

**Definition A.2** (Agnostic Learning). An algorithm $\mathcal{A}$ is said to be an $(\alpha, \beta)$-agnostic learner for concept class $\mathcal{H}$ with sample complexity $n$ if for any distribution $P$ over $\mathcal{X} \times \{0, 1\}$, it takes as input a dataset $S = ((x_1, y_1), \ldots, (x_n, y_n))$, where every $(x_i, y_i)$ is drawn i.i.d. from $P$, and outputs a hypothesis $h$ satisfying

$$\Pr[\mathrm{err}_P(h) \leq \inf_{h^\star \in \mathcal{H}} \mathrm{err}_P(h^\star) + \alpha] \geq 1 - \beta,$$

where the probability is taken over the random generation of $S$ and the random coins of $\mathcal{A}$.

A learner $\mathcal{A}$ is said to be *proper* if it always output some $h \in \mathcal{H}$. Otherwise we say $\mathcal{A}$ is *improper*.

**Definition A.3** (Growth Function). Let $S = (x_1, \ldots, x_n)$ be an unlabeled dataset. The projection of $\mathcal{H}$ onto $S$ is defined as

$$\Pi_{\mathcal{H}}(S) = \{((x_1, h(x_1)), \ldots, (x_n, h(x_n))) : h \in \mathcal{H}\}.$$

The growth function of $\mathcal{H}$ is defined as $\Pi_{\mathcal{H}}(n) = \max_{S \in \mathcal{X}^n} |\Pi_{\mathcal{H}}(S)|$.

Now we can define the Vapnik-Chervonenkis (VC) dimension [Vapnik and Chervonenkis, 1971], which characterizes the PAC and agnostic learnability of a concept class.

**Definition A.4.** Let $\mathcal{H}$ be a concept class over $\mathcal{X}$. The VC dimension of $\mathcal{H}$, denoted by $\mathrm{VCdim}(\mathcal{H})$, is the largest $d$ such that $\Pi_{\mathcal{H}}(d) = 2^d$.

Also, one can view $\mathcal{X}$ as the concept class and $\mathcal{H}$ as the domain. The dual VC dimension of $\mathcal{H}$ is then defined as $\mathrm{VCdim}^\star(\mathcal{H}) = \mathrm{VCdim}(\mathcal{X})$.

The following Sauer's lemma [Sauer, 1972] states that the growth function is polynomially bounded as long as the class has finite VC dimension.

**Lemma A.5** (Sauer's Lemma). *Let $\mathcal{H}$ be a concept class with VC dimension $d_V$. Then $\Pi_{\mathcal{H}}(n) \leq 2^n$ for any $n \leq d_V$ and*

$$\Pi_{\mathcal{H}}(n) \leq \sum_{i=0}^{d_V} \binom{n}{i} \leq \left(\frac{en}{d_V}\right)^{d_V}$$

*for all $n > d_V$.*

In this work, we may use the following technical lemma together with Sauer's lemma to derive sample complexity bounds.

**Lemma A.6** ([Shalev-Shwartz and Ben-David, 2014]). *Let $a \geq 1$ and $b > 0$. Then:*

$$x \geq 4a\ln(2a) + 2b \Rightarrow x \geq a\ln(x) + b.$$

The following realizable generalization result [Vapnik and Chervonenkis, 1971, Blumer et al., 1989] suggests that given sufficient examples, with high probability every pair of concepts with a small empirical disagreement also has a small generalization disagreement.

**Lemma A.7** (Realizable Generalization Bound). *Let $\mathcal{H}$ be a concept class with VC dimension $d_V$ and $P$ be a distribution over $\mathcal{X}$. Suppose $S \in \mathcal{X}^n$ is a dataset of size $n$, where each element in $S$ is drawn i.i.d. from $P$ and*

$$n \geq C\frac{d_V \ln(1/\alpha) + \ln(1/\beta)}{\alpha}$$

*for some universal constant $C$ (i.e., $C$ does not depend on $\mathcal{H}$ and $P$). Then with probability $1 - \beta$ over the random generation of $S$, we have for all $h_1, h_2 \in \mathcal{H}$:*

- *If $\mathrm{dis}_P(h_1, h_2) \leq \alpha$ then $\mathrm{dis}_S(h_1, h_2) \leq 2\alpha$.*

- *If $\mathrm{dis}_S(h_1, h_2) \leq \alpha$ then $\mathrm{dis}_P(h_1, h_2) \leq 2\alpha$.*

The above bound requires the disagreement to be small. This is in contrast to the following agnostic generalization bound, which provides an absolute upper bound on the difference between empirical error and generalization error. However, it incurs an extra factor of $1/\alpha$.

**Lemma A.8** (Agnostic Generalization Bound [Talagrand, 1994]). *Let $\mathcal{H}$ be a concept class with VC dimension $d_V$ and $P$ be a distribution over $\mathcal{X} \times \{0,1\}$. Suppose $S \in (\mathcal{X} \times \{0,1\})^n$ is a dataset of size $n$, where each element in $S$ is drawn i.i.d. from $P$ and*

$$n \geq C\frac{d_V + \ln(1/\beta)}{\alpha^2}$$

*for some universal constant $C$. Then with probability $1 - \beta$ over the random generation of $S$, we have $\sup_{h \in \mathcal{H}}|\mathrm{err}_S(h) - \mathrm{err}_P(h)| \leq \alpha$.*

In PAC (and agnostic) learning, the output hypothesis is required to have a low generalization error. In contrast, empirical learners produce hypotheses only with low empirical errors.

**Definition A.9** (Empirical Learner [Bun et al., 2015]). An algorithm $\mathcal{A}$ is said to be an $(\alpha, \beta)$-PAC empirical learner for concept class $\mathcal{H}$ with sample complexity $n$ if it takes as input a dataset $S \in (\mathcal{X} \times \{0,1\})^n$ such that $\min_{h^\star \in \mathcal{H}} \mathrm{err}_S(h^\star) = 0$, and outputs a hypothesis $h$ satisfying

$$\Pr[\mathrm{err}_S(h) \leq \alpha] \geq 1 - \beta.$$

Similarly, an algorithm $\mathcal{A}$ is said to be an $(\alpha, \beta)$-agnostic empirical learner for concept class $\mathcal{H}$ with sample complexity $n$ if it takes as input a dataset $S \in (\mathcal{X} \times \{0,1\})^n$ and outputs a hypothesis $h$ satisfying

$$\Pr[\mathrm{err}_S(h) \leq \min_{h^\star \in \mathcal{H}} \mathrm{err}_S(h^\star) + \alpha] \geq 1 - \beta.$$

When there are no privacy constraints, empirical learners can be trivially constructed. The following lemma shows that one can create private empirical learners from private learners.

**Lemma A.10** ([Li et al., 2025], Based on [Bun et al., 2015]). *Let $\mathcal{A}$ be an $(\varepsilon, \delta)$-differentially private $(\alpha, \beta)$-PAC learner for $\mathcal{H}$ with sample complexity $n$, where $\varepsilon \leq 1$ and $n \geq 1/\varepsilon$. Then there exists a $(1, O(\delta/\varepsilon))$-differentially private $(\alpha, \beta)$-PAC empirical learner $\mathcal{A}'$ for $\mathcal{H}$ with sample complexity $O(\varepsilon n)$. Moreover, if $\mathcal{A}$ is proper, then $\mathcal{A}'$ is also proper.*

*Remark.* A similar result can be derived for transforming agnostic learners to agnostic empirical learners [Bun et al., 2019]. However, this could be suboptimal in terms of $\alpha$ since agnostic learning requires $\Omega(\mathrm{VCdim}(\mathcal{H})/\alpha^2)$ examples [Simon, 1996] even without privacy.

## A.2 Concentration Inequalities

**Lemma A.11** (Hoeffding's Inequality [Hoeffding, 1963]). *Let $Z_1, \ldots, Z_n$ be independent bounded random variables with $Z_i \in [a, b]$. Then*

$$\Pr\left[\sum_{i=1}^{n} (\mathbb{E}[Z_i] - Z_i) \geq t\right] \leq \exp\left(-\frac{2t^2}{n(b-a)^2}\right)$$

*for all $t \geq 0$.*

**Lemma A.12** (Mcdiarmid's Inequality for Permutations [McDiarmid, 1989, Golowich and Livni, 2021, Talagrand, 1995, Costello, 2013]). *Suppose $f : \mathcal{Z}^n \to \mathbb{R}$ is some function such that $|f(\bar{z}_1, \ldots, \bar{z}_n) - f(\bar{z}'_1, \ldots, \bar{z}'_n)| \leq c$ for any two sequences $(\bar{z}_1, \ldots, \bar{z}_n)$ and $(\bar{z}'_1, \ldots, \bar{z}'_n)$ that differ in at most one element. Let $(z_1, \ldots, z_n) \in \mathcal{Z}^n$ be some fixed sequence and $\pi$ be a random permutation over $[n]$, then we have*

$$\Pr\left[\mathbb{E}[f(z_{\pi(1)}, \ldots, z_{\pi(n)})] - f(z_{\pi(1)}, \ldots, z_{\pi(n)}) \geq r\right] \leq \exp\left(-\frac{2r^2}{9nc^2}\right).$$

**Lemma A.13** (Chernoff Bound, Sampling Without Replacement [Chernoff, 1952, Hoeffding, 1963]). *Let $Z_1, \ldots, Z_n$ be random variables drawn without replacement from $(z_1, \ldots, z_N) \in \{0, 1\}^N$ ($N \geq n$) and $Z = \sum_{i=1}^{n} Z_i$ denote their sum. Then for any $t \in (0, 1)$, we have*

$$\Pr\left[Z \leq (1-t)\mathbb{E}[Z]\right] \leq \exp\left(-\frac{t^2\mathbb{E}[Z]}{2}\right).$$

**Lemma A.14** (Coupon Collector). *Let $X_1, \ldots, X_m$ be i.i.d. drawn from the uniform distribution over $[n]$. Suppose $m \geq 2n$ and $4\ln(1/\beta) \leq k \leq n$, then*

$$\Pr[|\{j \in [k] : \exists i \in [m], X_i = j\}| > k/2] \geq 1 - \beta.$$

*Proof.* For any $S \subseteq [k]$, we have

$$\Pr[\forall i \in [m], X_i \notin S] = \left(1 - \frac{|S|}{n}\right)^m \leq \exp(-m|S|/n).$$

Therefore,

$$
\begin{aligned}
&\Pr[|\{j \in [k] : \exists i \in [m], X_i = j\}| > k/2] \\
&= 1 - \Pr[\exists S \subseteq [k] \text{ with } |S| = \lceil k/2 \rceil \text{ s.t. } \forall i \in [m], X_i \notin S] \\
&\geq 1 - \binom{k}{\lceil k/2 \rceil} \exp(-m\lceil k/2 \rceil/n) \\
&\geq 1 - 2^k \exp(-k) \\
&\geq 1 - \beta.
\end{aligned}
$$

$\square$

## A.3 Closure Bounds Under Boolean Aggregation

The following notion of 0-covering number was introduced by Rakhlin et al. [2015].

**Definition A.15** ([Rakhlin et al., 2015]). Let $\mathcal{T}$ be an $\mathcal{X}$-valued tree of depth $n$ and $V$ be a set of $\{0, 1\}$-valued tree of depth $n$. We say $V$ is a 0-cover of $\mathcal{H}$ on $\mathcal{T}$ if for any $h \in \mathcal{H}$ and $y_1, \ldots, y_n \in \{0, 1\}^n$, there exists $\mathcal{V} \in V$ such that

$$\forall t \in [n], \ h(\mathcal{T}_t(y_1, \ldots, y_{t-1})) = \mathcal{V}_t(y_1, \ldots, y_{t-1}).$$

The 0-covering number of $\mathcal{H}$ on $\mathcal{T}$ is defined as

$$\mathcal{N}(0, \mathcal{H}, \mathcal{T}) = \min_{V \text{ is a 0-cover of } \mathcal{H} \text{ on } \mathcal{T}} |V|.$$

Also, define

$$\mathcal{N}(0, \mathcal{H}, n) = \max_{\mathcal{T} \text{ is an } \mathcal{X}\text{-valued tree of depth } n} \mathcal{N}(0, \mathcal{H}, \mathcal{T}).$$

They proved the following upper bound on the $0$-covering number for Littlestone classes, which can be seen as an analogy of the celebrated Sauer's lemma on trees.

**Lemma A.16** ([Rakhlin et al., 2015]). *Let $\mathcal{H}$ be a concept class with Littlestone dimension $d$. Then we have $\mathcal{N}(0, \mathcal{H}, n) \leq 2^n$ for any $n \leq d$ and*

$$\mathcal{N}(0, \mathcal{H}, n) \leq \sum_{i=0}^{d} \binom{n}{i} \leq \left(\frac{en}{d}\right)^d$$

*for all $n > d$.*

The following fact directly follows from the definition of shattering on trees (for a rigorous proof, see [Ghazi et al., 2021a]).

**Fact A.17.** *Let $\mathcal{H}$ be a concept class of Littlestone dimension $d$. Then $\mathcal{N}(0, \mathcal{H}, d) = 2^d$.*

Let $G : \{0,1\}^k \to \{0,1\}$ be a boolean function and $\mathcal{H}_1, \ldots, \mathcal{H}_k$ be $k$ hypothesis classes over domain $\mathcal{X}$. Define the hypothesis class $G(\mathcal{H}_1, \ldots, \mathcal{H}_k)$ as

$$G(\mathcal{H}_1, \ldots, \mathcal{H}_k) = \{G(h_1, \ldots, h_k) : h_1 \in \mathcal{H}_1, \ldots, h_k \in \mathcal{H}_k\},$$

where $G(h_1, \ldots, h_k)(x) = G(h_1(x), \ldots, h_k(x))$ for any $x \in \mathcal{X}$. The following lemma bounds the VC dimension and the Littlestone dimension of $G(\mathcal{H}_1, \ldots, \mathcal{H}_k)$. The upper bound on the VC dimension is by a classical argument of Dudley [1978] (see [Alon et al., 2020] for a detailed explanation) that leverages Sauer's lemma to bound the growth function. The upper bound on the Littlestone dimension is due to [Ghazi et al., 2021a] in a similar manner using Lemma A.16.

**Lemma A.18.** *Let $G : \{0,1\}^k \to \{0,1\}$ be a boolean function and $\mathcal{H}_1, \ldots, \mathcal{H}_k$ be $k$ hypotheses classes over domain $\mathcal{X}$. Let $d = \max_{i \in [k]} \mathrm{Ldim}(\mathcal{H}_i)$ and $d_V = \max_{i \in [k]} \mathrm{VCdim}(\mathcal{H}_i)$. Then we have*

$$\mathrm{Ldim}(G(\mathcal{H}_1, \ldots, \mathcal{H}_k)) = O(kd \log k)$$

*and*

$$\mathrm{VCdim}(G(\mathcal{H}_1, \ldots, \mathcal{H}_k)) = O(kd_V \log k).$$

An analogous argument also leads to the following bounds on the dual VC dimension and the dual Littlestone dimension. We include a proof for completeness.

**Lemma A.19.** *Let $G : \{0,1\}^k \to \{0,1\}$ be a boolean function and $\mathcal{H}_1, \ldots, \mathcal{H}_k$ be $k$ hypotheses classes over domain $\mathcal{X}$. Let $d^\star = \max_{i \in [k]} \mathrm{Ldim}^\star(\mathcal{H}_i)$ and $d_V^\star = \max_{i \in [k]} \mathrm{VCdim}^\star(\mathcal{H}_i)$. Then we have*

$$\mathrm{Ldim}^\star(G(\mathcal{H}_1, \ldots, \mathcal{H}_k)) = O(kd^\star \log k)$$

*and*

$$\mathrm{VCdim}^\star(G(\mathcal{H}_1, \ldots, \mathcal{H}_k)) = O(kd_V^\star \log k).$$

*Proof.* We bound the dual VC dimension first. Let

$$S = (G(h_1^1, \ldots, h_k^1), \ldots, G(h_1^n, \ldots, h_k^n))$$

be a dataset of size $n \geq d_V^\star$ over $G(\mathcal{H}_1, \ldots, \mathcal{H}_k)$. Construct $k$ datasets $S^1, \ldots, S^k$, where $S^i = (h_i^1, \ldots, h_i^n)$ is a dataset over $\mathcal{H}_i$ for every $i \in [k]$. By Sauer's lemma, we have (since the function $(en/x)^x$ is monotonically increasing when $1 \leq x \leq n$)

$$|\Pi_{\mathcal{X}}(S^i)| \leq \left(\frac{en}{\mathrm{VCdim}^\star(\mathcal{H}_i)}\right)^{\mathrm{VCdim}^\star(\mathcal{H}_i)} \leq \left(\frac{en}{d_V^\star}\right)^{d_V^\star}.$$

Then we can bound the size of projection of $\mathcal{X}$ onto $S$:

$$
\begin{aligned}
|\Pi_{\mathcal{X}}(S)| &= |\{(G(h_1^1, \ldots, h_k^1)(x), \ldots, G(h_1^n, \ldots, h_k^n)(x)) : x \in \mathcal{X}\}| \\
&= |\{(G(h_1^1(x), \ldots, h_k^1(x)), \ldots, G(h_1^n(x), \ldots, h_k^n(x))) : x \in \mathcal{X}\}| \\
&\leq |\{(G(h_1^1(x_1), \ldots, h_k^1(x_k)), \ldots, G(h_1^n(x_1), \ldots, h_k^n(x_k))) : (x_1, \ldots, x_k) \in \mathcal{X}^k\}| \\
&\leq |\{((h_1^1(x_1), \ldots, h_k^1(x_k)), \ldots, (h_1^n(x_1), \ldots, h_k^n(x_k))) : (x_1, \ldots, x_k) \in \mathcal{X}^k\}| \\
&= \Pi_{\mathcal{X}}(S^1) \times \cdots \times \Pi_{\mathcal{X}}(S^k) \\
&\leq (en/d_V^\star)^{kd_V^\star}.
\end{aligned}
$$

This implies $\Pi_{\mathcal{X}}(n) \leq (en/d_V^\star)^{kd_V^\star}$, which is $o(2^n)$ as $n \to \infty$. Therefore, $G(\mathcal{H}_1, \ldots, \mathcal{H}_k)$ has finite dual VC dimension. Denote it by $D_V^\star$, taking $n = D_V^\star$ gives $2^{D_V^\star} \leq (eD_V^\star/d_V^\star)^{kd_V^\star}$. Solving the inequality yields $D_V^\star = O(kd_V^\star \log k)$.

We now bound the dual Littlestone dimension in a similar way. Let $\mathcal{T}$ be a $G(\mathcal{H}_1, \ldots, \mathcal{H}_k)$-valued tree with depth $n \geq d^\star$. Then there exists $k$ trees $\mathcal{T}^1, \ldots, \mathcal{T}^k$ such that $\mathcal{T}^i$ is an $\mathcal{H}_i$-valued tree with depth $n$ for every $i \in [k]$, and

$$\mathcal{T}_t(y_1, \ldots, y_{t-1}) = G(\mathcal{T}_t^1(y_1, \ldots, y_{t-1}), \ldots, \mathcal{T}_t^k(y_1, \ldots, y_{t-1}))$$

for all $t \in [n]$ and $(y_1, \ldots, y_{t-1}) \in \{0,1\}^{t-1}$. By Lemma A.16, we have

$$\mathcal{N}(0, \mathcal{X}, \mathcal{T}^i) \leq \left(\frac{en}{\mathrm{Ldim}^\star(\mathcal{H}_i)}\right)^{\mathrm{Ldim}^\star(\mathcal{H}_i)} \leq \left(\frac{en}{d^\star}\right)^{d^\star}.$$

For every $i \in [k]$, pick a 0-cover $V^i = \{\mathcal{V}^{i,1}, \ldots, \mathcal{V}^{i,|V^i|}\}$ of $\mathcal{X}$ on $\mathcal{T}^i$ with size $|V^i| = \mathcal{N}(0, \mathcal{X}, \mathcal{T}^i)$. Construct
$$V = \{\mathcal{V}^{j_1, \ldots, j_k} : j_1 \in [|V^1|], \ldots, j_k \in [|V^k|]\},$$
where $\mathcal{V}^{j_1, \ldots, j_k}$ is a $\{0,1\}$-valued tree such that

$$\mathcal{V}_t^{j_1, \ldots, j_k}(y_1, \ldots, y_{t-1}) = G(\mathcal{V}_t^{1, j_1}(y_1, \ldots, y_{t-1}), \ldots, \mathcal{V}_t^{k, j_k}(y_1, \ldots, y_{t-1}))$$

for all $t \in [n]$ and $(y_1, \ldots, y_{t-1}) \in \{0,1\}^{t-1}$. Then we have $|V| \leq (en/d^\star)^{kd^\star}$. For any $x \in \mathcal{X}$ and $(y_1, \ldots, y_n) \in \{0,1\}^n$, for every $i \in [k]$ there exists $\mathcal{V}^{i,j_i} \in V^i$ such that

$$\forall t \in [n], \ x(\mathcal{T}_t^i(y_1, \ldots, y_{t-1})) = \mathcal{V}_t^{i,j_i}(y_1, \ldots, y_{t-1})$$

since $V^i$ is a 0-cover of $\mathcal{X}$ on $\mathcal{T}^i$. As a consequence, we have for all $t \in [n]$:

$$\begin{aligned}
x(\mathcal{T}_t(y_1, \ldots, y_{t-1})) &= G(x(\mathcal{T}_t^1(y_1, \ldots, y_{t-1})), \ldots, x(\mathcal{T}_t^k(y_1, \ldots, y_{t-1}))) \\
&= G(\mathcal{V}_t^{1,j_1}(y_1, \ldots, y_{t-1}), \ldots, \mathcal{V}_t^{k,j_k}(y_1, \ldots, y_{t-1})) \\
&= \mathcal{V}_t^{j_1, \ldots, j_k}(y_1, \ldots, y_{t-1}).
\end{aligned}$$

This means $V$ is a 0-cover of $\mathcal{X}$ on $\mathcal{T}$. The desired upper bound is then implied by the same calculation as for the dual VC dimension. $\qquad\square$

## A.4 Other Tools for Privacy

One of the basic mechanisms for ensuring differential privacy is the Laplace mechanism.

**Definition A.20** (Laplace Distribution). A random variable has probability distribution $\mathrm{Lap}(b)$ if its probability density function is $f(x) = \frac{1}{2b}\exp(-|x|/b)$.

**Definition A.21** (Sensitivity). Let $f : \mathcal{Z}^n \to \mathbb{R}$ be a function. We say $f$ has sensitivity $\Delta$ if for any neighboring datasets $S_1$ and $S_2$, we have $|f(S_1) - f(S_2)| \leq \Delta$.

**Lemma A.22** (Laplace Mechanism [Dwork et al., 2006b]). *Let $f$ be a function with sensitivity $\Delta$. The mechanism that takes as input a dataset $S \in \mathcal{Z}^n$ and outputs $f(S) + X$ with $X \sim \mathrm{Lap}(\Delta/\varepsilon)$ is $(\varepsilon, 0)$-differentially private. Moreover, we have*

$$\Pr_{X \sim \mathrm{Lap}(\Delta/\varepsilon)}\left[|X| \leq \frac{\ln(1/\beta)\Delta}{\varepsilon}\right] \geq 1 - \beta.$$

Given a finite set $R$ and a score function $q : \mathcal{Z}^n \times R \to \mathbb{R}$. We say $q$ has sensitivity $\Delta$ if $q(\cdot, h)$ has sensitivity $\Delta$ for all $h \in R$. The exponential mechanism takes as input a dataset $S \in \mathcal{Z}^n$ and selects an element $h \in R$ with probability

$$\frac{\exp(-\varepsilon \cdot q(S, h)/2\Delta)}{\sum_{f \in R} \exp(-\varepsilon \cdot q(S, f)/2\Delta)}.$$

**Lemma A.23** ([McSherry and Talwar, 2007]). *The exponential mechanism is $(\varepsilon, 0)$-differentially private. Moreover, with probability $1 - \beta$, it outputs an $h \in R$ such that*

$$q(S, h) \leq \min_{f \in R} q(S, f) + \frac{2\Delta}{\varepsilon}\ln(|R|/\beta).$$

# B  Proof of Theorem 3.1

We first prove the following important property of the subroutine Update.

**Lemma B.1.** *Let $\mathcal{H}$ be a concept class with Littlestone dimension $d$ and $\mathcal{F}$ be a subset of $\mathcal{H}$. Let $S_1^s, \ldots, S_{N_s}^s$ be the input of* Update *(Algorithm 1) such that $|S_i^s| \leq 2s$ for all $i \in [N_s]$. Define*

$$I_f^s = \{i \in [N_s/2] : S_{2i-1}^s \text{ and } S_{2i}^s \text{ are consistent with } f\}$$

*and $I_f^{s+1}$ similarly for the output $S_1^{s+1}, \ldots, S_{N_{s+1}}^{s+1}$.*

*Suppose for every $f \in \mathcal{F}$, we have*

$$\left|\{i \in I_f^s : \mathtt{SOA}(S_{2i-1}^s) \neq \mathtt{SOA}(S_{2i}^s)\}\right| \geq M.$$

*Then for any $0 < r_1 \leq \frac{M}{2} - 6$ and $r_2 > 0$, with probability at least*

$$1 - \left(\frac{e(4d+1)N_s}{2d}\right)^d \cdot \left(\exp\left(-\frac{2r_1^2}{M}\right) + \exp\left(-\frac{2r_2^2}{9N_{s+1}}\right)\right),$$

*it holds that for each $f \in \mathcal{F}$, either*

$$|\{i \in I_f^{s+1} : \mathtt{SOA}(S_{2i-1}^{s+1}) \neq \mathtt{SOA}(S_{2i}^{s+1})\}| > \frac{(M/2 - r_1)^2}{6N_{s+1}} - r_2,$$

*or there exists some $h_0$ (depends on $f$) such that*

$$|\{i \in I_f^{s+1} : \mathtt{SOA}(S_{2i-1}^{s+1}) = \mathtt{SOA}(S_{2i}^{s+1}) = h_0\}| > \frac{(M/2 - r_1)^2}{6N_{s+1}} - r_2.$$

*Proof.* Let $P$ be the set of unlabelled data points occurred in any $S_1^s, \ldots, S_{N_s}^s$, i.e.,

$$P = \bigcup_{i=1}^{N_s} \{x : (x, 0) \in S_i^s \vee (x, 1) \in S_i^s\}.$$

Let $Q = \{\bar{x}_i : i \in [N_{s+1}]\}$. Then we have $|P| \leq 2sN_s$ and $|Q| \leq N_s/2$. By Sauer's lemma, we can identify a subset $\mathcal{G} \subseteq \mathcal{F}$ with $|\mathcal{G}| \leq \left(\frac{em}{d}\right)^d$, where $m = (2d + 1/2)N_s \geq (2s + 1/2)N_s \geq |P \cup Q|$, such that for every $f \in \mathcal{F}$, there exists some $g \in \mathcal{G}$ such that $f$ and $g$ are consistent on both $P$ and $Q$. Hence, it suffices to first prove the conclusion for every $g \in \mathcal{G}$, then apply a union bound over $\mathcal{G}$.

Fix some $g \in \mathcal{G}$ and define the following set

$$U_g = \{i \in [N_{s+1}] : S_{\pi(i)}^{s+1} \text{ is consistent with } g\}.$$

By Hoeffding's inequality, we have

$$\Pr\left[|U_g| \leq M/2 - r_1\right] \leq \exp\left(-\frac{2r_1^2}{M}\right).$$

Note that according to our algorithm, $U_g$ only depends on the randomness of $\bar{y}_i$'s and is independent of $\pi$. Consequently, the above probability is only taken over the randomness of $\bar{y}_i$'s.

We then condition on a fixed set $U_g$ with $|U_g| > M/2 - r_1$. Let $c_h = |\{\mathtt{SOA}(S_{\pi(i)}^{s+1}) = h : i \in U_g\}|$ denote the number of occurrence of $h$ when running the $\mathtt{SOA}$ on $S_{\pi(i)}^{s+1}$ for all $i \in U_g$. As a consequence, $\sum_h c_h = |U_g|$. If $\max_h c_h < \frac{2|U_g|}{3}$, it follows that $\sum_h c_h^2 \leq \max_h c_h \sum_h c_h < \frac{2}{3}|U_g|^2$. Hence for

any $i \in [N_{s+1}/2]$, we have

$$
\Pr[i \in I_g^{s+1} \wedge \mathtt{SOA}(S_{2i-1}^{s+1}) \neq \mathtt{SOA}(S_{2i}^{s+1}) \mid U_g]
$$
$$
= \Pr[S_{2i-1}^{s+1} \text{ and } S_{2i}^{s+1} \text{ are consistent with } g \wedge \mathtt{SOA}(S_{2i-1}^{s+1}) \neq \mathtt{SOA}(S_{2i}^{s+1}) \mid U_g]
$$
$$
= \sum_h \frac{c_h}{N_{s+1}} \cdot \frac{\sum_{h' \neq h} c_{h'}}{N_{s+1} - 1}
$$
$$
= \frac{1}{N_{s+1}(N_{s+1} - 1)} \cdot \sum_h c_h(|U_g| - c_h)
$$
$$
= \frac{|U_g|^2 - \sum_h c_h^2}{N_{s+1}(N_{s+1} - 1)}
$$
$$
> \frac{|U_g|^2}{3N_{s+1}^2}.
$$

Therefore, we can leverage Mcdiarmid's inequality for permutations (by setting $f$ to be the function that counts $i \in [N_{s+1}/2]$ such that $i \in I_g^{s+1}$ and $\mathtt{SOA}(S_{2i-1}^{s+1}) \neq \mathtt{SOA}(S_{2i}^{s+1})$) to show

$$
\Pr[|\{i \in I_g^{s+1} : \mathtt{SOA}(S_{2i-1}^{s+1}) \neq \mathtt{SOA}(S_{2i}^{s+1})\}| \leq R - r_2 \mid U_g] \leq \exp\left(-\frac{2r_2^2}{9N_{s+1}}\right),
$$

where $R = \frac{(M/2 - r_1)^2}{6N_{s+1}} < N_{s+1}/2 \cdot \frac{|U_g|^2}{3N_{s+1}^2} < \mathbb{E}[f]$.

Now consider the case that $\max_h c_h \geq \frac{2|U_g|}{3}$. Let $h_0$ be the hypothesis such that $c_{h_0} = \max_h c_h$. Then for any $i \in [N_{s+1}/2]$, we have

$$
\Pr[S_{2i-1}^{s+1} \text{ and } S_{2i}^{s+1} \text{ are consistent with } g \wedge \mathtt{SOA}(S_{2i-1}^{s+1}) = \mathtt{SOA}(S_{2i}^{s+1}) = h_0 \mid U_g]
$$
$$
= \frac{c_{h_0}}{N_{s+1}} \cdot \frac{c_{h_0} - 1}{N_{s+1} - 1}
$$
$$
\geq \frac{4|U_g|^2 - 6|U_g|}{9N_{s+1}^2}
$$
$$
> \frac{|U_g|^2}{3N_{s+1}^2},
$$

where the last inequality is because $|U_g| > M/2 - r_1 \geq 6$. Similarly, we have

$$
\Pr[|\{i \in I_g^{s+1} : \mathtt{SOA}(S_{2i-1}^{s+1}) = \mathtt{SOA}(S_{2i}^{s+1}) = h_0\}| \leq R - r_2 \mid U_g] \leq \exp\left(-\frac{2r_2^2}{9N_{s+1}}\right).
$$

Putting what we have proved so far together, for every $g \in \mathcal{G}$, it holds with probability at least

$$
1 - \exp\left(-\frac{2r_1^2}{M}\right) - \exp\left(-\frac{2r_2^2}{9N_{s+1}}\right)
$$

that either

$$
|\{i \in I_g^{s+1} : \mathtt{SOA}(S_{2i-1}^{s+1}) \neq \mathtt{SOA}(S_{2i}^{s+1})\}| > R - r_2
$$

or there exists some $h_0$ such that

$$
|\{i \in I_g^{s+1} : \mathtt{SOA}(S_{2i-1}^{s+1}) = \mathtt{SOA}(S_{2i}^{s+1}) = h_0\}| > R - r_2.
$$

Applying a union bound over $\mathcal{G}$ yields the desired result. $\square$

We then analyze the privacy of Algorithm 2.

**Lemma B.2.** *Algorithm 2 is $(\varepsilon, \delta)$-differentially private.*

*Proof.* During the entire procedure, we run many instances of AboveThreshold on disjoint sequences. Therefore by Theorem 2.5, putting them all together is still $\varepsilon/2$-differentially private.

Now consider the multiple executions of PrivateHistogram. Note that changing a single input example $(x_t, y_t)$ only changes at most one $S_i^s$ for every $s$. Then by Theorem 2.6, each PrivateHistogram is $(\varepsilon/2d, \delta/d)$-differentially private. Since we run PrivateHistogram only once for every $s \in [d]$, basic composition ensures the overall algorithm is $(\varepsilon, \delta)$-differentially private.[3] $\qquad \square$

We now show the following utility guarantee using Lemma B.1. Combining Lemma B.2 and B.3 yields Theorem 3.1.

**Lemma B.3.** *Let $\mathcal{H}$ be a concept class with Littlestone dimension $d$ and $N_0 = 2^{\Theta(2^d)}(\ln(T/\beta) + \ln(1/\delta)/\varepsilon)$ be appropriately chosen. For any adaptive adversary generating the sequence $(x_1, y_1), \ldots, (x_T, y_T)$ in the realizable setting, Algorithm 2 makes at most*

$$O\left(\frac{2^{O(2^d)}(\log T + \log(1/\beta) + \log(1/\delta))}{\varepsilon}\right)$$

*mistakes with probability $1 - \beta$.*

*Proof.* First by Theorem 2.5 and the union bound, it holds with probability $1 - \beta/3$ that during the execution of each instance of AboveThreshold, the number of mistakes made by the algorithm is within $[\tau - \alpha, \tau + \alpha + 1]$, where $\tau$ is the threshold assigned to the instance and $\alpha = \frac{8(\ln T + \ln(6T/\beta))}{\varepsilon_0}$. In particular, if the instance was created with layer number $s$, the number of mistakes made during the execution is within

$$\left[N_s, N_s + \frac{16(\ln T + \ln(6T/\beta))}{\varepsilon_0} + 1\right].$$

We use $E_1$ to denote the above event.

Then by Theorem 2.6 and the union bound, it holds with probability 1 that

$$\sup_{h \in \mathcal{H}} |\overline{\text{Count}}_{V_s}(h) - \text{Count}_{V_s}(h)| \leq \frac{8d \ln(8d/\delta)}{\varepsilon_0}.$$

for every $s \in [d]$. We use $E_2$ to denote this event.

Moreover, consider an execution of AboveThreshold that eventually halts by returning $a_t = \top$. We know that the algorithm keeps outputting $h_t = h$ during the execution, where $h$ is the first element of $L_s$. Let $I \subseteq [N_s/2]$ be an index set with size $k \geq 4\ln(3T/\beta)$ and $I'$ be the collection of all $i_t$ (during the execution of this AboveThreshold) such that $h(x_t) \neq y_t$. By Lemma A.14, it holds with probability $1 - \beta/3T$ conditioned on $E_1$ that $|I \cap I'| \geq k/2$. Let $E_3$ be the event that this holds for all instances of AboveThreshold that terminates by returning $\top$. By the union bound, we have $\Pr[E_3 \mid E_1] \geq 1 - \beta/3$.

Let $\mathcal{F}_t$ be the set consisting of all $h \in \mathcal{H}$ that is consistent with the data points received up to round $t$. That is,
$$\mathcal{F}_t = \{h \in \mathcal{H} : h \text{ is consistent with } (x_1, y_1), \ldots, (x_t, y_t)\}.$$
Let $t_s$ denote the round at which $S_1^s, \ldots, S_{N_s}^s$ were created. Define $I_f^s(t)$ to be the set

$$\{i \in [N_s/2] : S_{2i-1}^s \text{ and } S_{2i}^s \text{ are consistent with } f\}$$

at the end of round $t$. For every $s \in \{0, 1, \ldots, d\}$, let $E_{4,s}$ denote the following event: for every $f \in \mathcal{F}_{t_s}$ either
$$|\{i \in I_f^s(t_s) : \text{SOA}(S_{2i-1}^s) \neq \text{SOA}(S_{2i}^s)\}| \geq M_s$$
or there exists some $h_0$ such that
$$|\{i \in I_f^s(t_s) : \text{SOA}(S_{2i-1}^s) = \text{SOA}(S_{2i}^s) = h_0\}| \geq M_s,$$
where $M_s = 128 \cdot 2^{-6 \cdot 2^s} N_s = 128 \cdot 2^{-6 \cdot 2^s} \cdot N_0 \cdot 2^{-s}$.

Since $S_1^0, \ldots, S_{N_0}^0$ are initialized as $\emptyset$, it follows that $I_f^s(t_0) = [N_0/2]$ and $\text{SOA}(S_{2i-1}^0) = \text{SOA}(S_{2i}^0) = \text{SOA}(\emptyset)$ for all $i \in [N_0/2]$. Therefore, $E_{4,0}$ happens with probability 1. We next bound

---

[3]The original statement of basic composition [Dwork et al., 2006a] is for static dataset only. In this work we actually use stronger versions of (basic and advanced) composition that work for adaptive inputs [Lyu, 2022, Vadhan and Wang, 2021, Henzinger et al., 2025].

the probability of $E_{4,s+1}$ conditioned on $E_1 \cup E_2 \cup E_3$ and $E_{4,s}$. If we set $N_0 \geq \frac{d \cdot 2^{6 \cdot 2^d + d} \ln(8d/\delta)}{\varepsilon_0}$, then

$$\frac{8d \ln(8d/\delta)}{\varepsilon_0} \leq 32 \cdot 2^{-6 \cdot 2^s} \cdot N_0 \cdot 2^{-s} = M_s/4.$$

By $E_2$, for every $f \in \mathcal{F}_{t_s}$ that satisfies the second property of $E_{4,s}$, we have $\overline{\mathrm{Count}}_{V_s}(h_0) \geq 3M_s/4$, which implies $h_0 \in L_s$. Observe that from round $t = t_s + 1$ to $t = t_{s+1}$ we only insert data points that are realizable by $\mathcal{F}_{t_{s+1}}$, it follows that $I_f^s(t_{s+1}) = I_f^s(t_s)$ for all $f \in \mathcal{F}_{t_{s+1}}$. Thus by $E_3$, we have for all $f \in \mathcal{F}_{t_{s+1}}$ that

$$|\{i \in I_f^s(t_{s+1}) : \mathtt{SOA}(S_{2i-1}^s) \neq \mathtt{SOA}(S_{2i}^s)\}| \geq M_s/2$$

if we set $k = M_s = 128 \cdot 2^{-6 \cdot 2^s} \cdot N_0 \cdot 2^{-s} \geq 4\ln(3T/\beta)$. Also, it is easy to see from our algorithm that $|S_i^s| \leq 2s$ for all $i \in [N_s]$ since we will at most insert one data point from the input and one from Update for every $s$. Now we have fulfilled the conditions of Lemma B.1. Setting $M = M_s/2$, $r_1 = M/4$ (this requires $r_1 = M_s/8 \leq M_s/4 - 6$, which can be satisfied by letting $N_0 \geq 2^{6 \cdot 2^d + d}$), and $r_2 = M^2/(384N_{s+1})$ gives

$$\frac{(M/2 - r_1)^2}{6N_{s+1}} - r_2 = \frac{M^2}{128N_{s+1}}$$

$$= \frac{M_s^2}{512N_{s+1}}$$

$$= \frac{16384 \cdot 2^{-12 \cdot 2^s} N_0^2 \cdot 2^{-2s}}{512N_0 \cdot 2^{-(s+1)}}$$

$$= 128 \cdot 2^{-6 \cdot 2^{s+1}} \cdot N_0 \cdot 2^{-(s+1)}$$

$$= M_{s+1}.$$

As a result, the event $E_{4,s+1}$ holds with probability

$$1 - \left(\frac{e(4d+1)N_s}{2d}\right)^d \cdot \left(\exp\left(-\frac{2r_1^2}{M}\right) + \exp\left(-\frac{2r_2^2}{9N_{s+1}}\right)\right)$$

$$\geq 1 - (7N_0)^d \left(\exp\left(-\frac{M_s}{16}\right) + \exp\left(\frac{M_s^4}{72 \cdot 384^2 N_{s+1}^3}\right)\right)$$

$$= 1 - (7N_0)^d \left(\exp\left(-\frac{8N_0}{2^{6 \cdot 2^s + s}}\right) + \exp\left(-\frac{2048N_0}{81 \cdot 2^{24 \cdot 2^s + s - 3}}\right)\right)$$

$$\geq 1 - 2\exp\left(-\frac{N_0}{2^{24 \cdot 2^d + d}} + d\ln(7N_0)\right)$$

$$\geq 1 - \frac{\beta}{3d}$$

conditioned on $E_1 \cup E_2 \cup E_3$ and $E_{4,s}$ as long as

$$2\exp\left(-\frac{N_0}{2^{24 \cdot 2^d + d}} + d\ln(7N_0)\right) \leq \frac{\beta}{3d} \Leftrightarrow N_0 \geq 2^{24 \cdot 2^d + d}\left(d\ln 7 + d\ln N_0 + \ln(6d/\beta)\right),$$

which, by Lemma A.6, can be established by requiring

$$N_0 \geq 4 \cdot 2^{24 \cdot 2^d + d} \cdot d\ln\left(2^{24 \cdot 2^d + d} \cdot 2d\right) + 2 \cdot 2^{24 \cdot 2^d + d}(d\ln 7 + \ln(6/\beta)).$$

Let $E_4 = E_{4,0} \cup \cdots \cup E_{4,d}$, we have $\Pr[E_4 \mid E_1 \cup E_2 \cup E_3] \geq 1 - \beta/3$.

Summarizing what we have proved so far gives $\Pr[E_1 \cup E_2 \cup E_3 \cup E_4] \geq 1 - \beta$ for some

$$N_0 = O\left(2^{O(2^d)}\left(\log(T/\beta) + \frac{\log(1/\delta)}{\varepsilon}\right)\right).$$

We now condition on $E_1 \cup E_2 \cup E_3 \cup E_4$ and bound the number of mistakes. Note that the size of $L_s$ at round $t = t_s$ can be bounded by

$$\frac{N_s/2}{3M_s/4 - M_s/4} \leq \frac{2^{6 \cdot 2^s}}{128}$$

for $s \in [d]$ and by $1 \leq 2^{6 \cdot 2^s}/128$ for $s = 0$. Hence, the number of mistakes before $s$ reaches $d$ is at most

$$\sum_{s=0}^{d-1} \left( N_s + \frac{16(\ln T + \ln(6T/\beta))}{\varepsilon_0} + 1 \right) \cdot \frac{2^{6 \cdot 2^s}}{128}$$

$$= O\left( \frac{2^{O(2^d)}(\log T + \log(1/\beta) + \log(1/\delta))}{\varepsilon} \right).$$

Once the value of $s$ reaches $d$, event $E_4$ indicates that $I_f^d(t_d) \geq M_d$ for all $f \in \mathcal{F}_{t_d}$. Notice that for every $i \in I_f^d(t_d)$, the SOA makes at least $d$ mistakes on $S_{2i-1}^d$ and $S_{2i}^d$. Therefore, the property that the SOA makes at most $d$ mistakes on any realizable sequence implies that $\text{SOA}(S_{2i-1}^d) = \text{SOA}(S_{2i}^d) = f$. Then by $E_2$, we have $f \in L_d$. This means we can make at most

$$\left( N_d + \frac{16(\ln T + \ln(6T/\beta))}{\varepsilon_0} + 1 \right) \cdot \frac{2^{6 \cdot 2^d}}{128} = O\left( \frac{2^{O(2^d)}(\log T + \log(1/\beta) + \log(1/\delta))}{\varepsilon} \right)$$

mistakes after $t = t_d$. Combining the two bounds yields the desired result. $\qquad\square$

## C    Proofs for Section 4.1

### C.1    Proof of Theorem 4.1

*Proof.* The privacy guarantee directly follows from the post-processing property of DP and the fact that we run $\mathcal{B}$ on disjoint batches. Let $E$ denote the event that all executions of $\mathcal{B}$ succeed, we have $\Pr[E] \geq 1 - T/B \cdot \beta$. In the rest of the proof we condition on $E$. Note that under event $E$, the input sequence is always a valid synthetic sequence. Thus, the algorithm won't fail.

Assume without loss of generality $T \equiv 0 \pmod{B}$. Consider the $b$-th batch and fix $S_1^b, \ldots, S_B^b$. Since $\mathcal{A}$ is proper, the utility guarantee of $\mathcal{B}$ gives (note that the error rate is $2\alpha$ since we require $\mathcal{B}$ to output a sanitized dataset, see our discussion after Definition 2.7)

$$\mathbb{E}\left[ \sum_{t=(b-1)B+1}^{bB} \mathbb{I}[h_t(x_t) \neq y_t] \right] = \sum_{t=(b-1)B+1}^{bB} \frac{1}{B} \sum_{i=1}^{B} \mathbb{E}_{f \sim \mathcal{A}_i(S_i^b)} \left[ \mathbb{I}[f(x_t) \neq y_t] \right]$$

$$= \frac{1}{B} \sum_{i=1}^{B} \mathbb{E}_{f \sim \mathcal{A}_i(S_i^b)} \left[ \sum_{t=(b-1)B+1}^{bB} \mathbb{I}[f(x_t) \neq y_t] \right]$$

$$\leq \frac{1}{B} \sum_{i=1}^{B} \mathbb{E}_{f \sim \mathcal{A}_i(S_i^b)} \left[ \sum_{t=(b-1)B+1}^{bB} \mathbb{I}[f(x_t') \neq y_t'] \right] + 2\alpha B.$$

Let $p_i^b$ be the probability that $\mathcal{A}(S_i^b)$ makes a mistake on the last element of $S_i^{b+1}$ (note that $p_i^b$ itself is a random variable). Since we perform a random permutation over the synthetic data sequence $((x_{(b-1)B+1}', y_{(b-1)B+1}'), \ldots, (x_{bB}', y_{bB}'))$, we have

$$\mathbb{E}[p_i^b] = \frac{1}{B} \mathbb{E}_{f \sim \mathcal{A}_i(S_i^b)} \left[ \sum_{t=(b-1)B+1}^{bB} \mathbb{I}[f(x_t') \neq y_t'] \right].$$

Summing over all batches yields

$$\mathbb{E}\left[ \sum_{t=1}^{T} \mathbb{I}[h_t(x_t) \neq y_t] \right] \leq \mathbb{E}\left[ \sum_{i=1}^{B} \sum_{b=1}^{T/B} p_i^b \right] + 2\alpha T.$$

Let $m_i(h)$ be the number of mistakes made by $h \in \mathcal{H}$ on $S_i^{T/B+1}$. Note that $S_1^{T/B+1}, \ldots, S_B^{T/B+1}$ are disjoint subsequences of $((x_1', y_1'), \ldots, (x_T', y_T'))$. We thus have

$$\mathbb{E}\left[\sum_{i=1}^{B}\sum_{b=1}^{T/B} p_i^b - \min_{h^\star \in \mathcal{H}}\sum_{t=1}^{T}[h^\star(x_t') \neq y_t']\right] \leq \mathbb{E}\left[\sum_{i=1}^{B}\sum_{b=1}^{T/B} p_i^b - \sum_{i=1}^{B}\min_{h^\star \in \mathcal{H}} m_i(h^\star)\right]$$

$$= \sum_{i=1}^{B}\mathbb{E}\left[\sum_{b=1}^{T/B} p_i^b - \min_{h^\star \in \mathcal{H}} m_i(h^\star)\right]$$

$$\leq B \cdot R(T/B),$$

where the last line is due to Lemma 4.1 of [Cesa-Bianchi and Lugosi, 2006] (see also Lemma 11 of [Gonen et al., 2019]) and the regret bound of $\mathcal{A}$. Again by the utility guarantee of $\mathcal{B}$ we have

$$\mathbb{E}\left[\min_{h^\star \in \mathcal{H}}\sum_{t=1}^{T}\mathbb{I}[h^\star(x_t) \neq y_t]\right] \geq \mathbb{E}\left[\min_{h^\star \in \mathcal{H}}\sum_{t=1}^{T}\mathbb{I}[h^\star(x_t') \neq y_t']\right] - 2\alpha T.$$

Hence, the overall expected regret can be bounded by

$$\mathbb{E}\left[\sum_{t=1}^{T}\mathbb{I}[h_t(x_t) \neq y_t] - \min_{h^\star \in \mathcal{H}}\sum_{t=1}^{T}\mathbb{I}[h^\star(x_t) \neq y_t]\right]$$

$$\leq \mathbb{E}\left[\sum_{i=1}^{B}\sum_{b=1}^{T/B} p_i^b - \min_{h^\star \in \mathcal{H}}\sum_{t=1}^{T}\mathbb{I}[h^\star(x_t') \neq y_t']\right] + 4\alpha T$$

$$\leq B \cdot R(T/B) + 4\alpha T.$$

Moreover, since every $h_t$ is produced by $\mathcal{A}$, Algorithm 3 is also proper. $\square$

### C.2   Proof of Corollary 4.3

*Proof.* By Theorem 4.2 and Lemma 2.8, there exists an $(\varepsilon, \delta)$-differentially private $(\alpha, \beta)$-sanitizer for $\mathcal{H}^{\text{label}}$ with sample complexity $\tilde{O}(d^6\sqrt{d^\star}/\varepsilon\alpha^2)$. For any $B \leq T$, this translates to a sanitizer with $\alpha = \tilde{O}(d^3\sqrt[4]{d^\star}/\sqrt{\varepsilon B})$ and $\beta = 1/T^2$. Then by Theorem 4.1 and the regret bound of proper online learner [Hanneke et al., 2021, Alon et al., 2021], we obtain a private online learner with expected regret $\tilde{O}(\sqrt{dTB} + Td^3\sqrt[4]{d^\star}/\sqrt{\varepsilon B})$. Choosing $B = \tilde{\Theta}((Td^5\sqrt{d^\star}/\varepsilon)^{1/2})$ gives the desired result. $\square$

## D   Sanitization with Better Sample Complexity

In this section, we discuss how to reduce the sample complexity of the sanitizer in [Ghazi et al., 2021b] by a factor of $1/\alpha$. We achieve this improvement in two steps: first refine the sample complexity of the private proper PAC learner of Ghazi et al. [2021b], then make it applicable in the framework of Bousquet et al. [2020].

### D.1   Refined Sample Complexity for Proper PAC Learning

The proof in [Ghazi et al., 2021b] utilizes the uniform convergence result (aka agnostic generalization) to ensure that the empirical error of every $\tilde{f} \in \tilde{\mathcal{F}}$ ($\tilde{\mathcal{F}}$ is some hypothesis class whose VC dimension is bounded by the Littlestone dimension of the given concept class $\mathcal{H}$) is close to its generalization error. This is indeed an overkill — their proof only requires this to hold for hypotheses with low error. Therefore, one could replace the uniform convergence bound by the following relative uniform convergence results.

**Lemma D.1** (Relative Uniform Convergence [Anthony and Bartlett, 1999, Anthony and Shawe-Taylor, 1993])**.** *Suppose $\mathcal{H}$ is a concept class over $\mathcal{X}$ with VC dimension $d_V$ and $P$ is a distribution over*

$\mathcal{X} \times \{0, 1\}$. *Let $S$ be a dataset of size $n$ where every data point in $S$ is drawn i.i.d. from $P^n$. For any $0 < \lambda, \mu < 1$, we have*

$$\Pr[\exists h \in \mathcal{H}, \mathrm{err}_P(h) > (1+\lambda)\mathrm{err}_S(h) + \mu] \leq 4\left(\frac{2en}{d_V}\right)^{d_V} \exp\left(\frac{-\lambda\mu n}{4(\lambda+1)}\right)$$

*and*

$$\Pr[\exists h \in \mathcal{H}, \mathrm{err}_S(h) > (1+\lambda)\mathrm{err}_P(h) + \mu] \leq 4\left(\frac{2en}{d_V}\right)^{d_V} \exp\left(\frac{-\lambda\mu n}{4(\lambda+1)}\right).$$

Such a modification leads to the following result, which saves a factor of $1/\alpha$.

**Theorem D.2** ([Ghazi et al., 2021b], Slightly Strengthened)**.** *Let $\mathcal{H}$ be a concept class with Littlestone dimension $d$. Then there exists an $(\varepsilon, \delta)$-differentially private proper $(\alpha, \beta)$-PAC learner for $\mathcal{H}$ with sample complexity $\tilde{O}\left(d^6/\varepsilon\alpha\right)$.*

### D.2 Sanitization via Proper PAC Learning

We now demonstrate how to construct a sanitizer using a proper PAC learner based on the sequential-fooling framework of [Bousquet et al., 2020]. The framework can be described as a sequential game played between a generator and a discriminator, where the discriminator holds a dataset $S$ and the generator wants to obtain an accurate sanitization of $S$. At each round $t$, the generator proposes a distribution $P_t$. The generator wins the game if $P_t$ and $\hat{P}_S$ are within error $\alpha$ with respect to $\mathcal{H}$. Otherwise, the discriminator returns some $h \in \mathcal{H}$ such that $|P(h) - \hat{P}_S(h)| > \alpha$. Bousquet et al. [2020] proved that the generator can always win the game within $\tilde{O}(d^\star/\alpha^2)$ rounds. They also showed how to simulate the discriminator using a private proper agnostic learner. Putting the two pieces together yields an algorithm for sanitization. In our construction, we will use the same generator and modify the discriminator so that a proper PAC learner can be directly employed.

We first leverage a technique from [Li et al., 2025] to construct a private agnostic empirical learner directly using a private PAC learner without incurring a generalization cost of $\tilde{O}(\mathrm{VCdim}(\mathcal{H})/\alpha^2)$.

**Lemma D.3.** *Suppose there is an $(\varepsilon, \delta)$-differentially private proper $(\alpha, \beta)$-PAC learner for $\mathcal{H}$ with sample complexity $m$. Then there exists an $(O(\varepsilon), O(\delta))$-differentially private proper $(O(\alpha), O(\beta))$-agnostic empirical learner for $\mathcal{H}$ with sample complexity*

$$n = O\left(m + \frac{d_V \log(1/\alpha) + \log(1/\beta)}{\varepsilon\alpha}\right),$$

*where $d_V$ is the VC dimension of $\mathcal{H}$.*

Applying the above lemma to the learner in Theorem D.2 leads to the following private proper agnostic empirical learner.

**Corollary D.4.** *Let $\mathcal{H}$ be a concept class with Littlestone dimension $d$. Then there exist an $(\varepsilon, \delta)$-differentially private proper $(\alpha, \beta)$-agnostic empirical learner for $\mathcal{H}$ with sample complexity $\tilde{O}(d^6/\varepsilon\alpha)$.*

We now prove Lemma D.3. Given a dataset $S$ of size $n$ and an index set $I \subseteq [n]$, we write $S^I$ to denote the collection containing elements from $S$ with indices in $I$. For simplicity, we may abuse notation and write $\mathrm{dis}_S(h_1, h_2)$ for labeled dataset $S$. This means we ignore the labels and only calculate the disagreement on the feature portion of $S$. We illustrate the conversion in Algorithm 4.

The following claim states the privacy guarantee of Algorithm 4. The proof is nearly identical to the proof of Lemma 15 in [Li et al., 2025].

**Claim D.5.** *Suppose $\mathcal{A}$ is $(1, \delta)$-differentially private and $1/n \leq \varepsilon \leq 0.1$. Then Algorithm 4 is $(O(\varepsilon), O(\varepsilon\delta))$-differentially private.*

*Proof.* Let $S_1$ and $S_2$ be two neighboring datasets and $O$ be any subset of outputs. Without loss of generality, we assume $S_1$ and $S_2$ differ in their first element, i.e., $S_1 = ((x_1, y_1), (x_2, y_2), \ldots, (x_n, y_n))$ and $S_2 = ((x'_1, y'_1), (x_2, y_2), \ldots, (x_n, y_n))$. Let $\mathcal{B}$ denote Algorithm 4. For any $I \subseteq [n]$ of size $m = \lceil \varepsilon n \rceil$ and $k \in \{1, 2\}$, define

$$p_k(I) = \Pr[\mathcal{B}(S_k) \in O \mid \text{the sampled indexed set is } I].$$

---

**Algorithm 4:** Agnostic empirical learner

---

**Global Parameter:** concept class $\mathcal{H}$, parameter $\varepsilon$
**Input:** private empirical PAC learner $\mathcal{A}$ for $\mathcal{H}$, private dataset $S = ((x_1, y_1), \ldots, (x_n, y_n))$
1 Sample $I \subseteq [n]$ of size $|I| = \lceil \varepsilon n \rceil$ uniformly at random.
2 Initialize $R = \emptyset$.
3 For every possible labeling in $\Pi_{\mathcal{H}}(S^I)$, add to $R$ an arbitrary $h \in \mathcal{H}$ that is consistent with the labeling.
4 Define $q(S, h) = \min_{f \in \mathcal{H}}\{\text{dis}_{S^I}(h, f) + \text{err}_S(f)\}$.
5 Choose $h_0 \in R$ using the exponential mechanism with privacy parameter $\varepsilon$, score function $q$, and sensitivity parameter $\Delta = 1/n$.
6 Let $D$ be the dataset constructed by relabeling $S^I$ with $h_0$.
7 Output $\mathcal{A}(D)$.

---

Since $I$ is sampled uniformly, we have

$$\Pr[\mathcal{B}(S_k) \in O] = \frac{1}{\binom{n}{m}} \sum_{I \in Q} p_k(I),$$

where $Q = \{I \subseteq [n] : |I| = m\}$.

Fix an index set $I$ and consider two cases: $1 \in I$ and $1 \notin I$. If $1 \notin I$, then $\mathcal{B}(S_1)$ and $\mathcal{B}(S_2)$ will construct the same set $R$. For every $h \in R$, there is some $f_h \in \mathcal{H}$ such that $\text{dis}_{S_1^I}(h, f_h) + \text{err}_{S_1}(f_h) = q(S_1, h)$, then we have

$$q(S_2, h) = \min_{f \in \mathcal{H}}\{\text{dis}_{S_2^I}(h, f) + \text{err}_{S_2}(f)\}$$

$$\leq \text{dis}_{S_2^I}(h, f_h) + \text{err}_{S_2}(f_h)$$

$$\leq \text{dis}_{S_1^I}(h, f_h) + \text{err}_{S_1}(f_h) + 1/n$$

$$\leq q(S_1, h) + 1/n.$$

By symmetry, we also have $q(S_1, h) \leq q(S_2, h) + 1/n$ for every $h \in \mathcal{H}$. Let $E_k(h)$ denote event that the hypothesis $h_0$ chosen by the exponential mechanism on $S_k$ is $h$. It then follows by Lemma A.23 that

$$\Pr[E_1(h)] \leq e^{\varepsilon} \Pr[E_2(h)]$$

for any $h$. The post-processing property of DP immediately implies

$$p_1(I) \leq e^{\varepsilon} p_2(I).$$

Now suppose $1 \in I$. Fix some $i \notin I$ and let $J = (I \setminus \{1\}) \cup \{i\}$ and $K = I \cap J$. Since $1 \in I$, we have

$$S_1^I \cap S_2^J = S_1^K = S_2^K,$$

whose size is exactly $|K| = m - 1$. Let $R_1^I$ and $R_2^J$ be the set $R$ constructed from $S_1^I$ and $S_2^J$. Pick a finite set $U \subseteq \mathcal{H}$ such that for every labeling of $S_1^K = S_2^K$, there is exactly one $h \in U$ consistent with this labeling. For each $h \in U$, define

$$P_1^I(h) = \{h' \in R_1^I : \text{dis}_{S_1^K}(h, h') = 0\}$$

and

$$P_2^J(h) = \{h' \in R_2^J : \text{dis}_{S_2^K}(h, h') = 0\}.$$

Since the label set is $\{0, 1\}$, we have $1 \leq |P_1^I(h)|, |P_2^J(h)| \leq 2$. For any $h_1 \in P_1^I(h)$ and $h_2 \in P_2^J(h)$, let $q^I(S_1, h_1)$ be the score function calculated on $S_1$ with sampled index set $I$ and $q^J(S_2, h_2)$ that on $S_2$ with sampled index set $J$. Since there is some $f_{h_1} \in \mathcal{H}$ such that $\text{dis}_{S_1^I}(h_1, f_{h_1}) + \text{err}_{S_1}(f_{h_1}) = q^I(S_1, h_1)$, we have

$$q^J(S_2, h_2) = \min_{f \in \mathcal{H}}\{\text{dis}_{S_2^J}(h_2, f) + \text{err}_{S_2}(f)\}$$

$$\leq \text{dis}_{S_2^J}(h_2, f_{h_1}) + \text{err}_{S_2}(f_{h_1})$$

$$\leq \text{dis}_{S_1^I}(h_1, f_{h_1}) + 1/m + \text{err}_{S_1}(f_{h_1}) + 1/n$$

$$\leq q^I(S_1, h_1) + 1/n + 1/m,$$

where the third line is because $h_1$ and $h_2$ agree on $S_1^I \cap S_2^J$, which has size $m-1$. Since $(1/n)/2\Delta = 1/2$ and $(1/m)/2\Delta = n/2m \leq 1/2\varepsilon$, we have

$$\exp(-\varepsilon q^J(S_2, h_2)/2\Delta) \geq \exp(-\varepsilon(q^I(S_1, h_1) + 1/n + 1/m)/2\Delta)$$
$$\geq \exp(-\varepsilon q^I(S_1, h_1)/2\Delta) \cdot \exp(-\varepsilon(1/2 + 1/2\varepsilon))$$
$$\geq \exp(-\varepsilon q^I(S_1, h_1)/2\Delta) \cdot \exp(-1),$$

where in the last inequality is because $\varepsilon \leq 1$. By symmetry, we also have

$$\exp(-\varepsilon q^I(S_1, h_1)/2\Delta) \geq \exp(-\varepsilon q^J(S_2, h_2)/2\Delta) \cdot \exp(-1).$$

Then the fact that $1 \leq |P_1^I|, |P_2^J| \leq 2$ gives

$$\sum_{h_1 \in P_1^I(h)} \exp(-\varepsilon q^I(S_1, h_1)/2\Delta) \geq \frac{1}{2} \sum_{h_2 \in P_2^J(h)} \exp(-\varepsilon q^J(S_2, h_2)/2\Delta) \cdot \exp(-1).$$

Summing over all $h_1 \in R_1^I$ yields

$$\sum_{f \in R_1^I} \exp(-\varepsilon q^I(S_1, f)/2\Delta) = \sum_{h \in U} \sum_{h_1 \in P_1^I(h)} \exp(-\varepsilon q^I(S_1, h_1)/2\Delta)$$
$$\geq \sum_{h \in U} \frac{1}{2} \sum_{h_2 \in P_2^J(h)} \exp(-\varepsilon q^J(S_2, h_2)/2\Delta) \cdot \exp(-1)$$
$$= \frac{1}{2e} \sum_{f \in R_2^J} \exp(-\varepsilon q^J(S_2, f)/2\Delta).$$

Let $D_1^I(h_1)$ be the dataset obtained by relabeling $S_1^I$ with $h_1$ and $D_2^J(h_2)$ be the one obtained by relabeling $S_2^J$ with $h_2$. Recall that $h_1$ and $h_2$ agree on $S_1^I \cap S_2^J$, which has size $m-1$. We know that $D_1^I(h_1)$ and $D_2^J(h_2)$ are neighboring datasets. Let $E_1^I(h_1)$ be the event of choosing $h_1$ when running on $S_1$ with sampled index set $I$ and $E_2^J(h_2)$ be the event of choosing $h_2$ when running on $S_2$ with sampled index set $J$. Since $\mathcal{A}$ is $(1, \delta)$-differentially private, we have

$$\Pr[E_1^I(h_1)] \cdot \Pr[\mathcal{A}(D_1^I(h_1)) \in O]$$
$$= \frac{\exp(-\varepsilon q^I(S_1, h_1)/2\Delta)}{\sum_{f \in R_1^I} \exp(-\varepsilon q^I(S_1, f)/2\Delta)} \cdot \Pr[\mathcal{A}(D_1^I(h_1)) \in O]$$
$$\leq 2e^2 \cdot \frac{\exp(-\varepsilon q^J(S_2, h_2)/2\Delta)}{\sum_{f \in R_2^J} \exp(-\varepsilon q^J(S_2, f)/2\Delta)} \cdot (e \Pr[\mathcal{A}(D_2^J(h_2)) \in O] + \delta)$$
$$= 2e^2 \Pr[E_2^J(h_2)] \cdot (e \Pr[\mathcal{A}(D_2^J(h_2)) \in O] + \delta).$$

Then we have the following relation between $p_1(I)$ and $p_2(J)$:

$$p_1(I) = \sum_{h \in U} \sum_{h_1 \in P_1^I(h)} \Pr[E_1^I(h_1)] \cdot \Pr[\mathcal{A}(D_1^I(h_1)) \in O]$$
$$\leq \sum_{h \in U} 2 \sum_{h_2 \in P_2^J(h)} 2e^2 \Pr[E_2^J(h_2)] \cdot (e \Pr[\mathcal{A}(D_2^J(h_2)) \in O] + \delta)$$
$$= 4e^3 p_2(J) + 4e^2 \delta.$$

Note that $\sum_{I \in Q:1 \in I} \sum_{i \in [n] \setminus I} p_2((I \setminus \{1\}) \cup \{i\})$ counts every $p_2(J)$ ($J \in Q$ and $1 \notin J$) exactly $|I| = m$ times. Therefore, we have

$$\sum_{I \in Q:1 \in I} p_1(I) = \frac{1}{n-m} \sum_{I \in Q:1 \in I} \sum_{i \in [n] \setminus I} p_1(I)$$
$$\leq \frac{1}{n-m} \sum_{I \in Q:1 \in I} \sum_{i \in [n] \setminus I} \left(4e^3 p_2((I \setminus \{1\}) \cup \{i\}) + 4e^2 \delta\right)$$
$$= \frac{m}{n-m} \sum_{J \in Q:1 \notin J} \left(4e^3 p_2(J) + 4e^2 \delta\right)$$
$$\leq 24e^3 \varepsilon \sum_{J \in Q:1 \notin J} p_2(J) + 4e^2 \delta \cdot \binom{n-1}{m-1},$$

where the last line is due to

$$\frac{m}{n-m} \cdot |\{J \in Q : 1 \notin J\}| = \frac{m}{n-m} \cdot \binom{n-1}{m} = \binom{n-1}{m-1}$$

and

$$\frac{m}{n-m} = \frac{\lceil \varepsilon n \rceil}{n - \lceil \varepsilon n \rceil} \le \frac{2\varepsilon n}{n - 2\varepsilon n} \le 6\varepsilon$$

as long as $\varepsilon n \ge 1$ and $\varepsilon \le 1/3$. Finally, we have

$$
\begin{aligned}
\Pr[\mathcal{B}(S_1) \in O] &= \frac{1}{|Q|} \left( \sum_{I \in Q : 1 \notin I} p_1(I) + \sum_{I \in Q : 1 \in I} p_1(I) \right) \\
&\le \frac{1}{\binom{n}{m}} \left( e^\varepsilon \sum_{I \in Q : 1 \notin I} p_2(I) + 24e^3\varepsilon \sum_{J \in Q : 1 \notin J} p_2(J) + 4e^2\delta \cdot \binom{n-1}{m-1} \right) \\
&\le (e^\varepsilon + 24e^3\varepsilon) \frac{1}{\binom{n}{m}} \sum_{I \in Q} p_2(I) + 4e^2\delta \cdot \frac{m}{n} \\
&= e^{O(\varepsilon)} \Pr[\mathcal{B}(S_2) \in O] + O(\varepsilon\delta).
\end{aligned}
$$

$\square$

The utility guarantee of Algorithm 4 is shown in the following claim.

**Claim D.6.** *Suppose $\mathcal{A}$ is a proper $(\alpha, \beta)$-PAC empirical learner for $\mathcal{H}$ with sample complexity $m = \lceil \varepsilon n \rceil$. Then Algorithm 4 is a proper $(O(\alpha), O(\beta))$-agnostic empirical learner with sample complexity $n$ as long as $m \ge C(d_V \log(1/\alpha) + \log(1/\beta))/\alpha$ for some constant $C$ and $n \ge 1/\varepsilon$, where $d_V$ is the VC dimension of $\mathcal{H}$. In other words,*

$$n = O\left( \frac{m}{\varepsilon} + \frac{d_V \log(1/\alpha) + \log(1/\beta)}{\varepsilon\alpha} \right).$$

*Proof.* By Sauer's Lemma, we have $|R| \le (em/d_V)^{d_V}$. Then by Lemma A.23, with probability at least $1 - \beta$ the exponential mechanism chooses some $h_0$ such that

$$q(S, h_0) \le \min_{h \in R} q(S, h) + \frac{2}{n\varepsilon} \ln(|R|/\beta) \le \min_{h \in R} q(S, h) + \alpha$$

as long as

$$n \ge \frac{2}{\varepsilon\alpha} \left( \ln(1/\beta) + d_V \ln(em/d_V) \right).$$

Since $m = \lceil \varepsilon n \rceil \le 2\varepsilon n$, by Lemma A.6, the above holds if

$$m \ge C_1 \frac{d_V \log(1/\alpha) + \log(1/\beta)}{\alpha}$$

for some constant $C_1$. Let $h^\star = \operatorname{argmin}_{f \in \mathcal{H}} \operatorname{err}_S(f)$. There exists some $h \in R$ such that $\operatorname{dis}_{S^I}(h^\star, h) = 0$. For such $h$, we have $q(S, h) = \operatorname{err}_S(h^\star)$. This implies $q(S, h_0) \le \operatorname{err}_S(h^\star) + \alpha$, or equivalently, $\operatorname{dis}_{S^I}(h_0, f_0) + \operatorname{err}_S(f_0) \le \operatorname{err}_S(h^\star) + \alpha$ for some $f_0 \in \mathcal{H}$.

Since $\operatorname{dis}_{S^I}(h_0, f_0)$ is non-negative, we have $\operatorname{err}_S(f_0) \le \operatorname{err}_S(h^\star) + \alpha$. Also, we have $\operatorname{dis}_{S^I}(h_0, f_0) \le \alpha$ because $\operatorname{err}_S(f_0) \ge \operatorname{err}_S(h^\star)$. Then by Lemma A.7 and the fact that the Chernoff bound is more concentrated for sampling without replacement [Hoeffding, 1963], with probability at least $1 - \beta$ we have

$$\operatorname{dis}_{S^I}(h_1, h_2) \le \alpha \Rightarrow \operatorname{dis}_S(h_1, h_2) \le 2\alpha$$

simultaneously holds for all $h_1, h_2 \in \mathcal{H}$ as long as

$$m \ge C_2 \frac{d_V \log(1/\alpha) + \log(1/\beta)}{\alpha}$$

for some constant $C_2$. As a consequence, we have $\text{dis}_S(h_0, f_0) \leq 2\alpha$.

Since $D$ is labeled by $h_0$, with probability $1 - \beta$ the output $g = \mathcal{A}(D)$ satisfies that $\text{dis}_{S^I}(g, h_0) \leq \alpha$. Moreover, we have $g \in \mathcal{H}$ since $\mathcal{A}$ is proper. Hence, Algorithm 4 is proper and $\text{dis}_S(g, h_0) \leq 2\alpha$. By the triangle inequality and the union bound, we have

$$\begin{aligned} \text{err}_S(g) &\leq \text{err}_S(f_0) + \text{dis}_S(f_0, g) \\ &\leq \text{err}_S(h^\star) + \alpha + \text{dis}_S(f_0, h_0) + \text{dis}_S(h_0, g) \\ &\leq \text{err}_S(h^\star) + 5\alpha \end{aligned}$$

with probability at least $1 - 3\beta$. $\qquad\square$

*Proof of Lemma D.3.* By Lemma A.10, there is a $(1, O(\delta/\varepsilon))$-differentially private proper $(\alpha, \beta)$-PAC empirical learner $\mathcal{A}$ for $\mathcal{H}$ with sample complexity $O(\varepsilon m)$. Then by Claim D.5, Algorithm 4 is $(O(\varepsilon), O(\delta))$-differentially private. Moreover, by Claim D.6, Algorithm 4 is a proper $(O(\alpha), O(\beta))$-agnostic empirical learner with sample complexity

$$n = O\left(m + \frac{d_V \log(1/\alpha) + \log(1/\beta)}{\varepsilon \alpha}\right).$$

$\qquad\square$

We now show how to construct the discriminator using proper agnostic empirical learner in the following lemma.

**Lemma D.7.** *Given a dataset $S \in \mathcal{X}^n$ and a public distribution $P_t$. Suppose for any $\mathcal{F} \subseteq \mathcal{H} \cup (1 - \mathcal{H})$, there is an $(\varepsilon/3, \delta)$-differentially private proper $(\alpha/10, \beta/3)$-agnostic empirical learner for $\mathcal{F}$ with sample complexity $n$ and*

$$n \geq C \frac{\ln(1/\alpha\beta)}{\varepsilon\alpha}$$

*for some constant $C$. Then there exists an $(\varepsilon, \delta)$-differentially private algorithm such that with probability $1 - \beta$:*

- *If it outputs some $h \in \mathcal{H} \cup (1 - \mathcal{H})$ then $\hat{P}_S(h) - P_t(h) \geq \alpha/2$.*

- *If it outputs "WIN" then $|\hat{P}_S(h) - P_t(h)| \leq \alpha$ for all $h \in \mathcal{H}$.*

*Proof.* Let $k = \lceil 10/\alpha \rceil$ and define

$$\mathcal{F}_i = \{h \in \mathcal{H} \cup (1 - \mathcal{H}) : P_t(h) \in [(i-1)/k, i/k]\}$$

for all $i \in [k]$. Construct score function $q(S, i) = -(\max_{h \in \mathcal{F}_i} \hat{P}_S(h) - i/k)$. It is easy to verify that the sensitivity of $q$ is $1/n$. By Lemma A.23, running the exponential mechanism with privacy parameter $\varepsilon/3$ returns some $j \in [k]$ such that $q(S, j) \leq \min_{i \in [k]} q(S, i) + 2\ln(3k/\beta)/\varepsilon n$ with probability $1 - \beta/3$. This implies

$$\max_{h \in \mathcal{F}_j} \hat{P}_S(h) - j/k \geq \max_{i \in [k]} \left(\max_{h \in \mathcal{F}_i} \hat{P}_S(h) - i/k\right) - \alpha/10$$

as long as

$$n \geq \frac{20 \ln(60/\alpha\beta)}{\varepsilon\alpha}.$$

We then construct a dataset $S'$ by labeling all data points in $S$ with 1 and run the proper agnostic empirical learner for $\mathcal{F}_j$ on $S'$ to find some $h_0 \in F_j$ such that

$$\text{err}_{S'}(h_0) \leq \min_{h \in \mathcal{F}_j} \text{err}_{S'}(h) + \alpha/10$$

with probability $1 - \beta/3$. This is equivalent to $\hat{P}_S(h_0) \geq \max_{h \in \mathcal{F}_j} \hat{P}_S(h) - \alpha/10$. We output $h_0$ if $\hat{P}_S(h_0) + X - j/k \geq 3\alpha/5$ and output "WIN" otherwise, where $X \sim \text{Lap}(3/\varepsilon)$. Note that this step is $(\varepsilon/3, 0)$-differentially private, and we have

$$\Pr[|X| \leq \alpha/10] \geq \Pr[|X| \leq 3\ln(3/\beta)/\varepsilon n] \geq 1 - \beta/3$$

given that
$$n \geq \frac{30 \ln(3/\beta)}{\varepsilon \alpha}.$$

The privacy guarantee directly follows from basic composition. By the union bound, with probability $1 - \beta$, if we output $h_0$ then

$$\hat{P}_S(h_0) \geq j/k - X + 3\alpha/5 \geq P_t(h_0) - \alpha/10 + 3\alpha/5 \geq P_t(h_0) + \alpha/2.$$

Otherwise, for any $i \in [k]$ and $h \in \mathcal{F}_i$ we have

$$\hat{P}_S(h) - P_t(h) \leq \hat{P}_S(h) - (i-1)/k$$
$$\leq \left( \max_{f \in \mathcal{F}_i} \hat{P}_S(f) - i/k \right) + 1/k$$
$$\leq \left( \max_{f \in \mathcal{F}_j} \hat{P}_S(f) - j/k \right) + \alpha/10 + 1/k$$
$$\leq \hat{P}_S(h_0) + \alpha/10 - j/k + \alpha/10 + 1/k$$
$$< 3\alpha/5 - X + \alpha/10 + \alpha/10 + 1/k$$
$$\leq 3\alpha/5 + \alpha/10 + \alpha/10 + \alpha/10 + \alpha/10$$
$$\leq \alpha.$$

The desired conclusion holds since $\mathcal{H} \cup (1 - \mathcal{H})$ is symmetric. $\qquad\square$

The property of the generator used in [Bousquet et al., 2020] is described in the following lemma.

**Lemma D.8** ([Bousquet et al., 2020]). *Let $\mathcal{H}$ be a concept class with dual Littlestone dimension $d^\star$. Suppose there is a discriminator such that:*

- *If it outputs some $h \in \mathcal{H} \cup (1 - \mathcal{H})$ then $\hat{P}_S(h) - P_t(h) \geq \alpha/2$.*

- *If it outputs "WIN" then $|\hat{P}_S(h) - P_t(h)| \leq \alpha$ for all $h \in \mathcal{H}$.*

*Then there exists a generator that makes the discriminator respond "WIN" within $O\left( \frac{d^\star}{\alpha^2} \log\left( \frac{d^\star}{\alpha} \right) \right)$ rounds.*

Following the proof strategy of [Ghazi et al., 2021b], which strengthens the proof of [Bousquet et al., 2020] by applying the advanced composition theorem, we are able to show Theorem 4.2.

*Proof of Theorem 4.2.* We have $\mathrm{Ldim}(\mathcal{H} \cup (1 - \mathcal{H})) = O(d)$ and $\mathrm{Ldim}^\star(\mathcal{H} \cup (1 - \mathcal{H})) = d^\star$ (see, e.g., [Alon et al., 2020] and [Bousquet et al., 2020]). By Corollary D.4, for any $\mathcal{F} \subseteq \mathcal{H} \cup (1 - \mathcal{H})$ there is an $(\varepsilon'/3, \delta')$-differentially private proper $(\alpha/10, \beta'/3)$-agnostic empirical learner for $\mathcal{F}$ with sample complexity $\tilde{O}(d^6/\varepsilon'\alpha)$. Then we can use Lemma D.7 to construct an $(\varepsilon', \delta')$-differentially private discriminator, which with probability $1 - \beta'$ either outputs "WIN" (hence, $|\hat{P}_S(h) - P_t(h)| \leq \alpha$ for all $h \in \mathcal{H}$) or some $h \in \mathcal{H} \cup (1 - \mathcal{H})$ such that $\hat{P}_S(h) - P_t(h) \geq \alpha/2$. Now run the generator in Lemma D.8. By setting $\beta' = \beta/T$, we know that with probability $1 - \beta$ the generator produces some $P_t$ such that $|\hat{P}_S(h) - P_t(h)| \leq \alpha$ for all $h \in \mathcal{H}$. To ensure the entire process is $(\varepsilon, \delta)$-differentially private, by advanced composition [Dwork et al., 2010b] it suffices to set

$$\delta' = \delta/2T \quad \text{and} \quad \varepsilon' = \frac{\varepsilon}{2\sqrt{2T \ln(2/\delta)}}.$$

Hence, the overall sample complexity is $\tilde{O}(d^6 \sqrt{T}/\varepsilon\alpha) = \tilde{O}(d^6 \sqrt{d^\star}/\varepsilon\alpha^2)$

$\qquad\square$

# E  Online Learning via Privately Constructing Experts

## E.1  Realizable Sanitization

We first define the notion of realizable sanitization.

**Definition E.1** (Realizable Sanitization). We say an algorithm is an $(\alpha, \beta)$-realizable sanitizer for $\mathcal{H}$ with input size $n$ if it takes as input a dataset $S \in \mathcal{X}^n$ and outputs a function $\text{Est} : \mathcal{H} \to \{0, 1\}$ such that with probability $1 - \beta$, for any $h \in \mathcal{H}$:

- If $\hat{P}_S(h) \geq \alpha$ then $\text{Est}(h) = 1$.

- If $\hat{P}_S(h) = 0$ then $\text{Est}(h) = 0$.

It turns out that we can again leverage private proper agnostic empirical learners to construct a private realizable sanitizer. The following lemma shows that we can construct a discriminator.

**Lemma E.2.** *Given a private dataset $S \in \mathcal{X}^n$ and a public function $Q_t : \mathcal{H} \to \{0, 1\}$. Suppose for any $\mathcal{F} \subseteq \mathcal{H}$ there is an $(\varepsilon/4, \delta/2)$-differentially private proper $(\alpha/9, \beta/4)$-agnostic empirical learner for $\mathcal{F}$ with sample complexity $n$ and*

$$n \geq \frac{36 \ln(4/\beta)}{\varepsilon \alpha}.$$

*Then there exists an $(\varepsilon, \delta)$-differentially private algorithm such that with probability $1 - \beta$:*

- *If it outputs $(h, 0)$ for some $h \in \mathcal{H}$ then*

$$\hat{P}_S(h) \leq 4\alpha/9 \quad and \quad Q_t(h) = 1.$$

- *If it outputs $(h, 1)$ for some $h \in \mathcal{H}$ then*

$$\hat{P}_S(h) \geq 5\alpha/9 \quad and \quad Q_t(h) = 0.$$

- *If it outputs "WIN" then for all $h \in \mathcal{H}$:*

$$\hat{P}_S(h) \geq \alpha \Rightarrow Q_t(h) = 1 \quad and \quad \hat{P}_S(h) = 0 \Rightarrow Q_t(h) = 0.$$

*Proof.* Let $\mathcal{H}_0 = \{h \in \mathcal{H} : Q_t(h) = 0\}$ and $S_0$ be the dataset obtained by labeling all data in $S$ with 1. We run an $(\varepsilon/4, \delta/2)$-differentially private proper $(\alpha/9, \beta/4)$-agnostic empirical learner for $\mathcal{H}_0$ on $S_0$ and obtain $h_0 \in \mathcal{H}$. With probability $1 - \beta/4$ we have

$$\text{err}_{S_0}(h_0) \leq \min_{h \in \mathcal{H}_0} \text{err}_{S_0}(h) + \alpha/9.$$

This is equivalent to $\hat{P}_S(h_0) \geq \max_{h \in \mathcal{H}_0} \hat{P}_S(h) - \alpha/9$. We output $(h_0, 1)$ and exit if $\hat{P}_S(h_0) + X_0 \geq 2\alpha/3$, where $X_0 \sim \text{Lap}(4/\varepsilon n)$. Note that this step is $(\varepsilon/4, 0)$-differentially private, and we have

$$\Pr[|X_0| \leq \alpha/9] \geq \Pr[|X_0| \leq 4\ln(4/\beta)/\varepsilon n] \geq 1 - \beta/4.$$

If we do not exit, then similarly let $\mathcal{H}_1 = \{h \in \mathcal{H} : Q_t(h) = 1\}$ and $S_1$ be the dataset obtained by labeling all data in $S$ with 0. We run an $(\varepsilon/4, \delta/2)$-differentially private proper $(\alpha/9, \beta/4)$-agnostic empirical learner for $\mathcal{H}_1$ on $S_1$ and obtain $h_1 \in \mathcal{H}$. With probability $1 - \beta/4$ we have

$$\text{err}_{S_1}(h_1) \leq \min_{h \in \mathcal{H}_1} \text{err}_{S_1}(h) + \alpha/9.$$

This is equivalent to $\hat{P}_S(h_1) \leq \min_{h \in \mathcal{H}_1} \hat{P}_S(h) + \alpha/9$. We output $(h_1, 0)$ if $\hat{P}_S(h_1) + X_1 \leq \alpha/3$, where $X_1 \sim \text{Lap}(4/\varepsilon n)$. Otherwise we output "WIN". Also, we have $\Pr[|X_1| \leq \alpha/9] \geq 1 - \beta/4$.

The privacy guarantee directly follows from basic composition. By the union bound, with probability $1 - \beta$ we have

$$\hat{P}_S(h_0) \geq 2\alpha/3 - X_0 \geq 2\alpha/3 - \alpha/9 = 5\alpha/9$$

if we output $(h_0, 1)$. If we instead output $(h_1, 0)$, then

$$\hat{P}_S(h_1) \leq \alpha/3 - X_1 \leq \alpha/3 + \alpha/9 = 4\alpha/9.$$

Otherwise if we output "WIN", we have

$$\begin{aligned}
\hat{P}_S(h) &\leq \hat{P}_S(h_0) + \alpha/9 \\
&< 2\alpha/3 - X_0 + \alpha/9 \\
&\leq 2\alpha/3 + \alpha/9 + \alpha/9 \\
&< \alpha
\end{aligned}$$

for all $h \in \mathcal{H}_0$ and

$$
\begin{aligned}
\hat{P}_S(h) &\geq \hat{P}_S(h_1) - \alpha/9 \\
&> \alpha/3 - X_1 - \alpha/9 \\
&\geq \alpha/3 - \alpha/9 - \alpha/9 \\
&> 0
\end{aligned}
$$

for all $h \in \mathcal{H}_1$ as desired. □

Given a concept class $\mathcal{H}$ over $\mathcal{X}$, we define a hypothesis class

$$
\mathcal{X}_{m,\alpha/2} = \{(x_1, \ldots, x_m) \in \mathcal{X}^m\}
$$

over $\mathcal{H}$, where every predicate $(x_1, \ldots, x_m) \in \mathcal{X}_{m,\alpha/2}$ is defined as

$$
(x_1, \ldots, x_m)(h) = \mathbb{I}\left[\frac{1}{m}\sum_{i=1}^{m} h(x_i) \geq \frac{\alpha}{2}\right].
$$

The right-hand side can be seen as a boolean function of $h(x_1), \ldots, h(x_m)$. Then Lemma A.18 provides an upper bound on the Littlestone dimension of $X_{m,\alpha/2}$.

**Claim E.3.** *Let $\mathcal{H}$ be a concept class over $\mathcal{X}$ with dual Littlestone dimension $d^\star$. Then the Littlestone dimension of $\mathcal{X}_{m,\alpha/2}$ is at most $O(md^\star \log m)$.*

It can be shown that for any dataset $S$, the class $\mathcal{X}_{m,\alpha/2}$ contains a good realizable sanitization of $S$ as long as $m$ is sufficiently large.

**Lemma E.4.** *Let $\mathcal{H}$ be a concept class over $\mathcal{X}$ with VC dimension $d_V$. Set $m = Cd_V \ln(1/\alpha)/\alpha$, where $C$ is some universal constant. For any dataset $S$ over $\mathcal{X}$, there exists $(x_1, \ldots, x_m) \in \mathcal{X}^m$ such that for all $h \in \mathcal{H}$*

- *If $\hat{P}_S(h) \geq 5\alpha/9$ then $\frac{1}{m}\sum_{i=1}^{m} h(x_i) \geq \alpha/2$.*

- *If $\hat{P}_S(h) \leq 4\alpha/9$ then $\frac{1}{m}\sum_{i=1}^{m} h(x_i) < \alpha/2$.*

*Proof.* Let $x_1, \ldots, x_m$ be i.i.d. drawn from $\hat{P}_S$. By Lemma A.7 (or Lemma D.1), the desired property holds with probability $1/2$. Hence there exists a realization with the property. □

Given the above result, one can obtain a realizable sanitization by using any online learner (e.g., the SOA) to interact with the discriminator from Lemma E.2. This leads to the following theorem.

**Theorem E.5.** *Let $\mathcal{H}$ be a concept class over $\mathcal{X}$ with Littlestone dimension $d$, dual Littlestone $d^\star$, and VC dimension $d_V$. Then there exists an $(\varepsilon, \delta)$-differentially private $(\alpha, \beta)$-realizable sanitizer for $\mathcal{H}$ with sample complexity*

$$
\tilde{O}\left(\frac{d^6\sqrt{d^\star d_V}}{\varepsilon\alpha^{1.5}}\right).
$$

*Proof.* Set $m$ as in Lemma E.4. Let $D$ be the Littlestone dimension of $\mathcal{X}_{m,\alpha/2}$ and $T = D + 1$. By Claim E.3, we have $T = O(md^\star \log m)$. Run a sequential game using the SOA for $\mathcal{X}_{m,\alpha/2}$ as the generator and the algorithm from Lemma E.2 with privacy parameter $(\varepsilon', \delta')$ and success probability $1 - \beta/T$ as the discriminator. By the union bound, with probability $1 - \beta$ the discriminator succeeds for all rounds. Condition on this event, if the discriminator outputs "WIN" at some round $t$ then we can simply output $\text{Est} = Q_t$ and exit.

Thus, it suffices to prove that the discriminator always outputs "WIN" at some round under the above event. Suppose, for the sake of contradiction, that it produces a sequence $((h_1, y_1), \ldots, (h_T, y_T))$. Then we have $Q_t(h_t) \neq y_t$ for all $t \in [T]$. By Lemma E.4, there exists some $Q = (x_1, \ldots, x_m) \in \mathcal{X}_{m,\alpha/2}$ such that for all $h \in \mathcal{H}$:

- If $\hat{P}_S(h) \geq 5\alpha/9$ then $Q(h) = 1$.

- If $\hat{P}_S(h) \leq 4\alpha/9$ then $Q(h) = 0$.

Consequently, we have $Q(h_t) = y_t$ for all $t \in [T]$. This means the entire sequence is realizable by $\mathcal{X}_{m,\alpha/2}$. However, the SOA makes $T = D + 1$ mistakes, a contradiction.

In order to ensure the entire process is $(\varepsilon, \delta)$-differentially private, by advanced composition [Dwork et al., 2010b], it suffices to set

$$\delta' = \delta/2T \quad \text{and} \quad \varepsilon' = \frac{\varepsilon}{2\sqrt{2T\ln(2/\delta)}}.$$

We now analyze the sample complexity. It depends on the sample complexity of the learner used in Lemma E.2. Employing the one in Corollary D.4 results in a sample complexity of

$$\tilde{O}\left(\frac{d^6}{\varepsilon'\alpha}\right) = \tilde{O}\left(\frac{d^6\sqrt{T}}{\varepsilon\alpha}\right) = \tilde{O}\left(\frac{d^6\sqrt{d^\star d_V}}{\varepsilon\alpha^{1.5}}\right)$$

$\square$

### E.2 Constructing Experts

We first extends our realizable sanitizer to the sequential setting by the binary mechanism [Dwork et al., 2010a, Chan et al., 2011], which is based on the following fact.

**Fact E.6.** *Let $T > 1$ be the time horizon. There exists a set $I \subseteq \{(l,r) : l, r \in [T] \text{ and } l \leq r\}$ and a universal constant $C$ such that:*

- $|I| \leq CT$.

- *For every $t \in [T]$, we have $|\{(l,r) \in I : l \leq t \leq r\}| \leq C\ln T$.*

- *For any $L, R \in [T]$ and $L \leq R$, there exists $L = t_1 < \cdots < t_u < t_{u+1} = R + 1$ for some $u \leq C\ln T$ such that $(t_i, t_{i+1} - 1) \in I$ for every $i \in [u]$.*

We describe the sequential realizable sanitizer in Algorithm 5. Note that it actually sanitizes a larger hypothesis class $\mathcal{H}_{m,1/2} \oplus \mathcal{H}$, where

$$\mathcal{H}_{m,1/2} = \{(h_1, \ldots, h_m) \in \mathcal{H}^m\}$$

is a hypothesis class over $\mathcal{X}$ and the predicate $(h_1, \ldots, h_m)$ is defined as

$$(h_1, \ldots, h_m)(x) = \mathbb{I}\left[\frac{1}{m}\sum_{i=1}^{m} h_i(x) \geq \frac{1}{2}\right].$$

Since the right-hand side of the above is a boolean function and the operation $\oplus$ is also a boolean function, the following claim then follows from Lemma A.18 and A.19.

**Claim E.7.** *Let $\mathcal{H}$ be a concept class with Littlestone dimension $d$, dual Littlestone dimension $d^\star$, and VC dimension $d_V$. Then $\mathcal{H}_{m,1/2} \oplus \mathcal{H}$ has Littlestone dimension $O(md\log m)$, dual Littlestone dimension $O(md^\star \log m)$, and VC dimension $O(md_V \log m)$.*

In Algorithm 5, the algorithm $\mathcal{B}$ is indeed a series of realizable sanitizers with varying error parameters $\alpha(n)$ for different input sizes $n$ because we have to sanitize sequences with different lengths. Furthermore, we assume for simplicity that an $(\alpha(n), \beta)$-realizable sanitizer directly outputs a synthetic dataset of size $n$ rather than an estimation. This is done by first computing Est, then finding a dataset $S'$ of size $n$ such that $\hat{P}_{S'}(f) > 0$ for all $f \in \mathcal{H}_{m,1/2} \oplus \mathcal{H}$ with $\text{Est}(f) = 1$. Note that with probability $1 - \beta$, the private dataset $S$ satisfies this property and hence such $S'$ exists. The lemma below summarizes the privacy and utility properties of Algorithm 5.

**Lemma E.8.** *Let $\mathcal{B}$ be an $(\varepsilon, \delta)$-differentially private $(\alpha(n), \beta)$-realizable sanitizer for $\mathcal{H}_{m,1/2} \oplus \mathcal{H}$ with input size $n$ and $C$ be the constant in Fact E.6. Then Algorithm 5 is $(C\ln T \cdot \varepsilon, C\ln T \cdot \delta)$-differentially private. Moreover, let $\Delta = \max_{n \in [T]} n\alpha(n)$. Then with probability $1 - CT\beta$, for all $1 \leq l \leq r \leq T$ we have*

$$(r - l + 1)\hat{P}_{S_{l,r}}(f) \geq C\ln T\Delta \Rightarrow \hat{P}_{S'_{l,r}}(f) > 0$$

*for all $f \in \mathcal{H}_{m,1/2} \oplus \mathcal{H}$, where $S_{l,r} = (x_l, \ldots, x_r)$.*

*Proof.* The privacy guarantee directly follows from Fact E.6 and basic composition. Since $|I| \leq CT$, with probability at least $1 - CT\beta$ all executions of $\mathcal{B}$ succeed. For any $1 \leq l \leq r \leq T$, we have $S'_{l,r} = (S'_{t_1,t_2-1}, \ldots, S'_{t_u,t_{u+1}-1})$ for some $l = t_1 < \cdots < t_u < t_{u+1} = r + 1$ and $u \leq C \ln T$. Then for any $f \in \mathcal{H}_{m,1/2} \oplus \mathcal{H}$ such that $(r - l + 1)\hat{P}_{S_{l,r}}(f) \geq C \ln T \Delta$, there is some $i \in [u]$ such that $(t_{i+1} - t_i)\hat{P}_{S_{t_i,t_{i+1}-1}}(f) \geq \Delta \geq (t_{i+1} - t_i)\alpha(t_{i+1} - t_i)$. This implies $\hat{P}_{S'_{t_i,t_{i+1}-1}}(f) > 0$ (since running $\mathcal{B}$ on $S_{t_i,t_{i+1}-1}$ gives $\mathrm{Est}(f) = 1$) and hence $\hat{P}_{S'_{l,r}}(f) > 0$. $\qquad\square$

---

**Algorithm 5:** Sanitization for intervals

**Global Parameter:** time horizon $T$, hypothesis class $\mathcal{H}_{m,1/2} \oplus \mathcal{H}$
**Input:** realizable sanitizer $\mathcal{B}$ for $\mathcal{H}_{m,1/2} \oplus \mathcal{H}$, data sequence $(x_1, \ldots, x_T)$

1 Let $I$ be the set in Fact E.6.
2 **for** $t = 1, \ldots, T$ **do**
3     **for** $l = t, t - 1, \ldots, 1$ **do**
4         **if** $(l, t) \in I$ **then**
5             $S'_{l,t} \leftarrow \mathcal{B}(x_l, \ldots, x_t)$.
6         **else**
7             Find $l = t_1 < \cdots < t_u < t_{u+1} = t + 1$ for some $u \leq C \ln T$ such that $(t_i, t_{i+1} - 1) \in I$ for every $i \in [u]$ as in Fact E.6.
8             $S'_{l,t} \leftarrow (S'_{t_1,t_2-1}, \ldots, S'_{t_{u-1},t_u-1})$.
9         **end**
10     **end**
11 **end**

---

We now present our construction of experts. As illustrated in Algorithm 6, each expert is indexed by $i_1, \ldots, i_M, j_1, \ldots, j_M$ such that $1 \leq j_1 \leq i_1 < j_2 \leq i_2 < \cdots < j_M \leq i_M \leq T$. The expert will keep the output unchanged from round $i_k + 1$ to round $i_{k+1}$. After round $i_{k+1}$, it changes the output by feeding a sanitized data point $x'_{j_k}$ to the online learner $\mathcal{A}$ and forcing $\mathcal{A}$ to make a mistake. Note that for the expert, it suffices to receive $(x'_{i_k+1}, \ldots, x'_{i_{k+1}})$ at round $i_{k+1}$ rather than in a real-time manner.

---

**Algorithm 6:** Expert

**Global Parameter:** time horizon $T$, concept class $\mathcal{H}$
**Input:** online learner $\mathcal{A}$ for $\mathcal{H}$, indices $i_1, \ldots, i_M, j_1, \ldots, j_M$, data sequence $(x'_1, \ldots, x'_T)$

1 $S \leftarrow \emptyset$.
2 **for** $t = 1, \ldots, T$ **do**
3     Output $h_t = \mathcal{A}(S)$.
4     **if** $t = i_k$ *for some* $k \in [M]$ **then**
5         $S \leftarrow (S, (x'_{j_k}, 1 - \mathcal{A}(S)(x'_{j_k})))$.
6     **end**
7 **end**

---

As we discussed in Section 4.2, the structure of the classifiers outputted by the online learner $\mathcal{A}$ cannot be too complex. Otherwise we have to sanitize a huge hypothesis class and this may lead to an unacceptable error rate. Therefore, we exploit the online learner proposed by Hanneke et al. [2021] whose output hypothesis at each round is a sparse majority of concepts in $\mathcal{H}$ (i.e., in $\mathcal{H}_{m,1/2}$ for some $m$).

**Lemma E.9** ([Hanneke et al., 2021])**.** *Let $\mathcal{H}$ be a concept class with Littlestone dimension $d$ and dual VC dimension $d_V^\star$. There exists a realizable online learner whose output hypothesis at each round is always in $\mathcal{H}_{m,1/2}$ and has a mistake bound of $M = O(d)$, where $m = O(d_V^\star)$.*

**Theorem E.10.** *Let $\mathcal{H}$ be a concept class with Littlestone dimension $d$, dual Littlestone dimension $d^\star$, VC dimension $d_V$, and dual VC dimension $d_V^\star$. Then there exists an $(\varepsilon, \delta)$-differentially private algorithm that receives an adaptively generated data sequence $(x_1, \ldots, x_T)$ and constructs a set*

---

**Algorithm 7:** Constructing experts

---

    **Global Parameter:** time horizon $T$, concept class $\mathcal{H}$, hypothesis class $\mathcal{H}_{m,1/2}$

    **Input:** online learner $\mathcal{A}$ for $\mathcal{H}$ with output in $\mathcal{H}_{m,1/2}$, data sequence $(x_1, \ldots, x_T)$

**1**   $J \leftarrow \{(i_1, \ldots, i_M, j_1, \ldots, j_M) : 1 \le j_1 \le i_1 < j_2 \le i_2 < \cdots < j_M \le i_M \le T\}.$

**2**   Initialize $\mathsf{Expert}(i_1, \ldots, i_M, j_1, \ldots, j_M)$ for every $(i_1, \ldots, i_M, j_1, \ldots, j_M) \in J$.

**3**   Let $\mathcal{B}$ be Algorithm 5.

**4**   **for** $t = 1, \ldots, T$ **do**

**5**      Feed $x_t$ to $\mathcal{B}$ and receive $S'_{1,t}, \ldots, S'_{t,t}$ from $\mathcal{B}$.

**6**      **foreach** $(i_1, \ldots, i_M, j_1, \ldots, j_M) \in J$ **do**

**7**          Receive $h_t^{(i_1, \ldots, i_M, j_1, \ldots, j_M)}$ from $\mathsf{Expert}(i_1, \ldots, i_M, j_1, \ldots, j_M)$.

**8**          **if** $t = i_k$ *for some* $k \in [M]$ **then**

**9**              Feed $S'_{i_{k-1}+1, i_k}$ to $\mathsf{Expert}(i_1, \ldots, i_M, j_1, \ldots, j_M)$ (define $i_0 = 0$).

**10**          **end**

**11**      **end**

**12**   **end**

---

*of $N = O(T^{O(d)})$ experts such that with probability at least $1 - \beta$, for any $h \in \mathcal{H}$ there exists an expert with output $(h_1, \ldots, h_T)$ such that*

$$\sum_{t=1}^{T} \mathbb{I}[h_t(x_t) \ne h(x_t)] = \tilde{O}\left(T^{1/3} \frac{(d_V^\star)^{14/3} d^5 (d^\star d_V)^{1/3}}{\varepsilon^{2/3}}\right).$$

*Proof.* Write $S_{l,r} = (x_l, \ldots, x_r)$. By Claim E.7, $\mathcal{H}_{m,1/2} \oplus \mathcal{H}$ has Littlestone dimension $D = O(md \log m)$, dual Littlestone dimension $D^\star = O(md^\star \log m)$, and VC dimension $D_V = O(md_V \log m)$. By Theorem E.5, there exists an $(\varepsilon/C \ln T, \delta/C \ln T)$-differentially private $(\alpha(n), \beta/CT)$-realizable sanitizer for $\mathcal{H}_{m,1/2} \oplus \mathcal{H}$ with input size $n$, where

$$\alpha(n) = \left(\frac{D^6 \sqrt{D^\star D_V}}{\varepsilon n}\right)^{2/3}.$$

Then by Lemma E.8, running Algorithm 5 with this sanitizer is $(\varepsilon, \delta)$-differentially private and with probability $1 - \beta$ we have

$$(r - l + 1)\hat{P}_{S_{l,r}}(f) \ge C \ln T \Delta \Rightarrow \hat{P}_{S'_{l,r}}(f) > 0$$

for all $1 \le l \le r \le T$ and $f \in \mathcal{H}_{m,1/2} \oplus \mathcal{H}$, where $C$ is the constant in Fact E.6 and

$$\Delta = \max_{n \in [T]} n\alpha(n) = \tilde{O}\left(T^{1/3} \cdot \left(\frac{D^6 \sqrt{D^\star D_V}}{\varepsilon}\right)^{2/3}\right).$$

Set $m$ and $M$ as in Lemma E.9 and use the online learner to construct Expert (Algorithm 6). Running Algorithm 7 gives a set of experts with size $N = |J| = O(T^M) = O(T^{O(d)})$. Since the algorithm can be seen as post-processing of the output of Algorithm 5. The overall algorithm is also $(\varepsilon, \delta)$-differentially private. Now suppose there exists some $h \in \mathcal{H}$ such that

$$\sum_{t=1}^{T} \mathbb{I}[h_t^{i_1, \ldots, i_M, j_1, \ldots, j_M}(x_t) \ne h(x_t)] > \lceil C \ln T \Delta \rceil \cdot M$$

for every $(i_1, \ldots, i_M, j_1, \ldots, j_M) \in J$. Note that $\mathsf{Expert}(i_1, \ldots, i_M, j_1, \ldots, j_M)$ only changes its output after round $t = i_1$, then there is some $i'_1$ such that

$$\sum_{t=1}^{i'_1} \mathbb{I}[h_t^{i'_1, i_2 \ldots, i_M, j_1, \ldots, j_M}(x_t) \ne h(x_t)] = \lceil C \ln T \Delta \rceil \ge C \ln T \Delta$$

for every $(i'_1, i_2, \ldots, i_M, j_1, \ldots, j_M) \in J$. This implies $\hat{P}_{S'_{1,i'_1}}(f \oplus h) > 0$ for

$$f = h_1^{i'_1, i_2 \ldots, i_M, j_1, \ldots, j_M} = \cdots = h_{i'_1}^{i'_1, i_2 \ldots, i_M, j_1, \ldots, j_M}.$$

Let $S'_{1,i'_1} = (x'_1, \ldots, x'_{i'_1})$. There is some $j'_1 \in [i'_1]$ such that

$$h^{i'_1, i_2 \ldots, i_M, j'_1, j_2, \ldots, j_M}_{j'_1}(x_{j'_1}) \neq h(x_{j'_1}).$$

By induction, we can similarly identify $i'_2, j'_2, \ldots, i'_M, j'_M$ such that

$$\sum_{t=i'_{k-1}+1}^{i'_k} \mathbb{I}[h^{i'_1, \ldots, i'_M, j'_1, \ldots, j'_M}_t(x_t) \neq h(x_t)] = \lceil C \ln T\Delta \rceil$$

and

$$h^{i'_1, \ldots, i'_M, j'_1, \ldots, j'_M}_{j'_k}(x_{j'_k}) \neq h(x_{j'_k})$$

for every $k \in [M]$. This means until round $t = i'_M$, the online learner $\mathcal{A}$ used by Expert$(i'_1, \ldots, i'_M, j'_1, \ldots, j'_M)$ received a sequence of length $M$ that is labeled by $h$ and made a mistake on every data point. However, by our assumption, we have

$$\sum_{t=i'_M+1}^{T} \mathbb{I}[h^{i'_1, \ldots, i'_M, j'_1, \ldots, j'_M}_t(x_t) \neq h(x_t)] > 0.$$

This contradicts Lemma E.9. Hence, for all $h \in \mathcal{H}$, there exists $(i_1, \ldots, i_M, j_1, \ldots, j_M) \in J$ such that

$$\sum_{t=1}^{T} \mathbb{I}[h^{i_1, \ldots, i_M, j_1, \ldots, j_M}_t(x_t) \neq h(x_t)] = O(\log T\Delta M)$$

$$= \tilde{O}\left(T^{1/3}\left(\frac{m^6 d^6 \sqrt{m^2 d^\star d_V}}{\varepsilon}\right)^{2/3} \cdot d\right)$$

$$= \tilde{O}\left(T^{1/3}\left(\frac{(d^\star_V)^7 d^6 \sqrt{d^\star d_V}}{\varepsilon}\right)^{2/3} \cdot d\right).$$

$\square$

### E.3 Incorporating Private OPE

Now we can run any private OPE algorithm over the experts constructed in Theorem E.10. We use the following results from [Jain and Thakurta, 2014] and [Asi et al., 2024] for adaptive and oblivious adversaries, respectively.

**Theorem E.11** ([Jain and Thakurta, 2014]). *For the OPE problem with $N$ experts, there exists an $(\varepsilon, \delta)$-differentially private algorithm with an expected regret of*

$$O\left(\frac{\sqrt{T\log(1/\delta)}\log N}{\varepsilon}\right)$$

*against any adaptive adversary.*

**Theorem E.12** ([Asi et al., 2024]). *For the OPE problem with $N$ experts, there exists an $(\varepsilon, \delta)$-differentially private algorithm with an expected regret of*

$$O\left(\sqrt{T\log N} + \frac{T^{1/3}\log N \log(T/\delta)}{\varepsilon^{2/3}}\right).$$

*against any oblivious adversary.*

Putting the above results and Theorem E.10 together yields the following regret bounds.

**Corollary E.13.** *Let $\mathcal{H}$ be a concept class with Littlestone dimension $d$, dual Littlestone dimension $d^\star$, VC dimension $d_V$, and dual VC dimension $d^\star_V$. Then there exists an $(\varepsilon, \delta)$-differentially private online learner for $\mathcal{H}$ with an expected regret of*

$$O\left(\frac{\sqrt{T\log(1/\delta)}d\log T}{\varepsilon}\right) + \tilde{O}\left(T^{1/3} \cdot \frac{(d^\star_V)^{14/3}d^5(d^\star d_V)^{1/3}}{\varepsilon^{2/3}}\right)$$

*against any adaptive adversary. Moreover, if the adversary is oblivious, then there exists an $(\varepsilon, \delta)$-differentially private learner for $\mathcal{H}$ with an expected regret of*

$$O\left(\sqrt{dT \log T}\right) + \tilde{O}\left(T^{1/3} \cdot \frac{(d_V^\star)^{14/3} d^5 (d^\star d_V)^{1/3}}{\varepsilon^{2/3}}\right).$$

