# OpenReview forum: "Private Online Learning against an Adaptive Adversary: Realizable and Agnostic Settings"
_NeurIPS.cc/2025/Conference — NeurIPS 2025 poster_

### Official Review · Reviewer_r3aS · 2025-07-02

**Clarity:** 4
**Significance:** 4
**Originality:** 3
**Rating:** 5
**Confidence:** 3

**Summary:**

The authors study the problem of online learning with differential privacy. They make the following contributions:
1. In the realizable setting, they propose a differentially private algorithm that achieves a mistake bound of $O_d(\log T)$ for an adaptive adversary, where $d$ is the Littlestone dimension of the hypothesis class and $T$ is the time horizon. This improves upon the previously known bound of $O_d(\sqrt{T})$.
2. In the agnostic setting, they propose a differentially private algorithm that achieves a regret bound of $\tilde{O}_d(\sqrt{T})$. However, the proposed algorithm is improper. They also propose a proper learner with a regret bound of $\tilde{O}_d(T^{3/4})$.

**Questions:**

In many online learning scenarios, the time horizon $T$ is not known in advance. I noticed that several parameters—such as the threshold in Algorithm 2 and the number of experts in Section 4.2—depend explicitly on $T$. Is it possible to remove this dependency?

**Ethical Concerns:**

["NO or VERY MINOR ethics concerns only"]

**Final Justification:**

The authors have answered my question adequately, and after going through their rebuttal, I have decided to keep my positive score.
Concerns resolved:
1. Doubling trick for unknown time horizon

**Limitations:**

yes

**Paper Formatting Concerns:**

N/A.

**Quality:**

4

**Strengths And Weaknesses:**

The paper is well written, with ideas presented in a clear and concise manner. The authors address the important problem of ensuring differential privacy in online learning, considering both the realizable and agnostic settings. While the work builds on prior research—such as Golowich and Livni [2021] and Bun et al. [2020]—and employs established tools like SOA, AboveThreshold, and Sanitization, it introduces several clever ideas and novel arguments (e.g., Lazy Update and uniform convergence) to improve upon the existing results.

---

> ### Author Rebuttal · Authors · 2025-07-31
>
> We thank the reviewer for the constructive comments, and our detailed responses are listed below.
>
> > In many online learning scenarios, the time horizon $T$ is not known in advance. I noticed that several parameters—such as the threshold in Algorithm 2 and the number of experts in Section 4.2—depend explicitly on $T$. Is it possible to remove this dependency?
>
> Yes, this dependency can be removed by the classical "doubling trick" -- splitting the input sequence into buckets of sizes $1,2,4,8,\cdots$ and running our algorithm on each bucket separately (with parameter $T=1,2,4,8,\cdots$). It can be shown that the resulting algorithm have the same regret bound while not requiring knowledge of the entire time horizon in advance [Cesa-Bianchi and Lugosi, 2006].
>
> We will include the above comment in the revision.
>
> Nicolo Cesa-Bianchi and Gábor Lugosi. Prediction, learning, and games. Cambridge University Press, 2006.

---

### Official Review · Reviewer_Nyfq · 2025-07-02

**Clarity:** 3
**Significance:** 3
**Originality:** 3
**Rating:** 5
**Confidence:** 3

**Summary:**

In online learning, a learner receives a sequence of $T$ data points from an adversary and has to respond at each time step with an hypothesis. A concept class $\mathcal{H}$ is said to be privately online learnable if there is an online learner achieving sublinear regret, that is also differentially-private w.r.t. all the stream of returned hypotheses. Apart from the classical differences, agnostic VS realizable setting and proper VS improper learner, one can differentiate between an oblivious adversary, that chooses the entire sequence of data points at the beginning, and adaptive one that can determine the sequence online, after observing the learner’s outputs. The authors show (1) an algorithm improving the mistake bound presented by Golowich and Livni [2021] against an adaptive adversary (closing the gap between the oblivious/adaptive cases) in the realizable setting and (2) an algorithm achieving sublinear regret for classes with finite Littlestone dimension in the agnostic setting with an adaptive adversary.

**Questions:**

1) Can you justify the necessity of including the condition over the dual Littlestone dimension in Theorem 4.2/ Corollary 4.3? This condition (if I am not mistaken) is required also for the results in "Sample-efficient proper PAC learning with approximate differential privacy" and I would like to have an intuitive explanation.
2) Do you believe it is possible to eliminate the $2^{2^{O(d)}}$  factor in the Littlestone dimension? Just, your opinion about it.

**Ethical Concerns:**

["NO or VERY MINOR ethics concerns only"]

**Final Justification:**

My opinion about the work after the rebuttal stays positive. This is a solid paper that addresses interesting questions regarding differentially private online learning.

**Limitations:**

yes

**Paper Formatting Concerns:**

I did not notice concerns of this kind.

**Quality:**

3

**Strengths And Weaknesses:**

**Positive points**:
The paper is generally well-written and easy to follow, even if quite dense. Studying differential privacy in online learning is an interesting topic and not generally an easy one. In Section 3, the realizable setting is analyzed: the authors first give an in-depth explanation of the reasons why the procedure by Golowich and Livni [2021] does not attain a regret that is logarithmic in $T$ and then propose an algorithm that efficiently addresses those limitations. I think that the analysis is interesting and sound and the algorithm makes use of known procedures as subroutines. In Section 4, the authors deal with the agnostic setting: in particular they show a simpler algorithm employing sanitization and a more complex one employing experts and sanitization and achieving a better regret bound with an improper learner.  It is interesting to see many techniques blended together to obtain those results.

**Negative points**:
As mentioned, the paper is dense. Without a knowledge of many results in the literature, it becomes a bit hard to follow some passages, even if the authors do their best to sum everything up. Also, I noticed that the main result cited at the beginning is not referenced again in Section 3 and that can be confusing.

**Minor comments**:
Typo in algorithm 2: failuare -> failure. Typo in Theorem 2.6 Pirvate -> Private.

All the above comments motivate my score.

---

> ### Author Rebuttal · Authors · 2025-07-31
>
> We thank the reviewer for the constructive comments and provide our detailed responses below.
>
> > Can you justify the necessity of including the condition over the dual Littlestone dimension in Theorem 4.2/ Corollary 4.3? This condition (if I am not mistaken) is required also for the results in "Sample-efficient proper PAC learning with approximate differential privacy" and I would like to have an intuitive explanation.
>
> Our derived bounds in Theorem 4.2/ Corollary 4.3 depend on the dual Littlestone dimension because our algorithm is based on sanitization. As proved in "Sample-efficient proper PAC learning with approximate differential privacy" (their Theorem 6.7), the sample complexity of sanitization grows at least polynomially in the dual Littlestone dimension. Below we provide an intuitive explanation for a special case of their result.
>
> Consider a domain $X=\\{0,1\\}^D$ and a class $H=\\{h_1,\dots,h_D\\}$ where $h_i(x) = x_i$. The Littlestone dimension of $H$ is $\log D$ and the dual Littlestone dimension is $D$. For this class, sanitizing a dataset is equivalent to privately estimating its $D$-dimensional mean, which requires $\Omega(\sqrt{D})$ samples (Steinke, T., & Ullman, J. (2015) ).
>
> Steinke, T., & Ullman, J. (2015). Between Pure and Approximate Differential Privacy. *ArXiv, abs/1501.06095*. Presented at the First Workshop on the Theory and Practice of Differential Privacy (TPDP ‘15), London, UK, April 2015.
>
> > Do you believe it is possible to eliminate the $2^{2^{O(d)}}$ factor in the Littlestone dimension? Just, your opinion about it.
>
> Yes, we believe that this factor can be improved to polynomial in $d$, as in "Sample-efficient proper PAC learning with approximate differential privacy" for PAC learning.
>
> > **Negative points**: As mentioned, the paper is dense. Without a knowledge of many results in the literature, it becomes a bit hard to follow some passages, even if the authors do their best to sum everything up. Also, I noticed that the main result cited at the beginning is not referenced again in Section 3 and that can be confusing.
> >
> > **Minor comments**: Typo in algorithm 2: failuare -> failure. Typo in Theorem 2.6 Pirvate -> Private.
>
> Thank you for pointing out these issues. We will fix the typos and reference our main result in Section 3 in the revision.

---

> > ### Comment · Reviewer_Nyfq · 2025-08-07
> >
> > I thank the authors for their response and clarifications.
> > I will maintain my score, as well.

---

### Official Review · Reviewer_9xHS · 2025-07-02

**Clarity:** 1
**Significance:** 4
**Originality:** 4
**Rating:** 4
**Confidence:** 3

**Summary:**

This paper addresses private online learning in the adaptive setting, where an adversary can choose inputs based on the learner's previous responses. The authors achieve two main results that improve upon existing mistake bounds.
1) Realizable Setting: The authors provide an algorithm with an O_d(log(T)) mistake bound for learning any concept class with Littlestone dimension d, which closes the gap to the known lower bound. Their approach builds on Golowich and Livni (2021) but introduces a key innovation: instead of releasing updates at every timestep (which accumulates privacy costs), they use "lazy updates" that defer computation until necessary, followed by random permutation to maintain the privacy analysis structure.
2) Agnostic Setting: The authors develop a new algorithm achieving an O_d(√(T)) mistake bound. Their approach first constructs a private proper learner, then applies sanitization techniques for generic Littlestone classes from Ghazi et al. (2021) to improve the regret bound, though this results in an improper learner.

**Questions:**

Can you explain why in Algorithm 2, you don’t need to reinitialize SOA after a lazy update?

**Ethical Concerns:**

["NO or VERY MINOR ethics concerns only"]

**Final Justification:**

The rebuttal has clarified my initial doubts, and provided the authors implement the suggested changes in their final version, I'm changing my original score to be more positive.

**Limitations:**

yes

**Quality:**

2

**Strengths And Weaknesses:**

Strengths:
The results are foundational in our understanding of private online learning in the adaptive setting and improve upon previous works in this area by giving better upper bounds in the realizable setting and new results in the agnostic setting.

Weaknesses:
The explanations in Sections 3 and 4 lack sufficient clarity and detail, which diminishes the correctness of the proposed techniques. The authors would benefit from reallocating space from the extensive Preliminaries section to provide a more detailed explanation of Algorithm 2 and its different components.
While I understand the high-level approach of addressing the √(T) composition cost at each timestep through lazy updates, several technical concerns remain unaddressed. In the adaptive setting, don’t you have to re-initialize the SOA algorithm after every lazy update? Otherwise, the shared randomness across lazy updates could introduce correlations between historical predictions and current outputs, potentially creating exploitable patterns for adversaries.
This concern suggests that the privacy analysis extends beyond straightforward composition arguments and involves more nuanced considerations. However, Section 3 does not appear to address these subtleties, leaving important questions about their solution unresolved.

---

> ### Author Rebuttal · Authors · 2025-07-31
>
> We thank the reviewer for the constructive comments and provide our detailed responses below.
>
> > The authors would benefit from reallocating space from the extensive Preliminaries section to provide a more detailed explanation of Algorithm 2 and its different components.
>
> Thank you for your suggestion, we will elaborate on the details of Algorithm 2 to better explain it in Section 3 in the revision.
>
> > don’t you have to re-initialize the SOA algorithm after every lazy update?
>
> We do not reinitialize the SOA. During the update, every $S_{i}^{s+1}$ is obtained by appending an example to some $S_{j}^s$ (as seen in line 7 of Algorithm 1). Hence, $SOA(S_{i}^{s+1})$ can be seen as feeding a new example to $SOA(S_{j}^s)$ rather than reinitializing it.
>
> > the shared randomness across lazy updates could introduce correlations between historical predictions and current outputs, potentially creating exploitable patterns for adversaries.
>
> We believe this sentence and its subsequent description in this review is a misunderstanding, possibly caused by our (overly) concise writing due to the page limit. We present more technical details in Appendix B. We are confident that there is no mistake in our derivation.
>
> While correlations exist between historical predictions and current outputs, they do not compromise the correctness of our algorithm.
>
> The correctness of our algorithm is based solely on the fact that a constant fraction of the sequences ($S_1^s,\dots,S_{N_s}^s$) remain consistent with the input (Lemma B.1). We prove this through concentration bounds only over the permutation $\pi$ and label $\bar{y}_i$ in the update (Algorithm 1). Since they are generated during the update and are independent of any prior randomness, the correctness can be directly established.
>
> We emphasize that shared randomness is crucial for achieving a small mistake bound -- without it, the only way to ensure privacy is by advanced composition (as in the work of Golowich and Livni (2021) ), which results in a $\sqrt{T}$ mistake bound. The primary challenge in private online learning is how to effectively leverage shared randomness to mitigate privacy cost while achieving a small mistake bound.
>
> > the privacy analysis extends beyond straightforward composition arguments and involves more nuanced considerations.
>
> The privacy analysis remains straightforward considering the following three facts:
>
> 1. the AboveThreshold mechanism is private against an adaptive adversary.
> 2. the PrivateHistogram is private and non-sequential (thus also private against an adaptive adversary).
> 3. the composition theorem holds for an adaptive adversary.
>
> Consequently, the entire algorithm is private against an adaptive adversary. Note that the randomness involved in the update is only for utility purpose and does not affect the privacy analysis.

---

### Official Review · Reviewer_2ZzJ · 2025-07-03

**Clarity:** 3
**Significance:** 3
**Originality:** 3
**Rating:** 5
**Confidence:** 4

**Summary:**

This submission studies differentially private online learning of Littlestone classes against an adaptive adversary. It builds on work of Golowich and Livni, who previously showed that classes with finite Littlestone dimension can be learned with mistake bound $O(\log T)$ against an oblivious adversary, and with mistake bound $O(\sqrt{T})$ against an adaptive adversary. This paper improves these results in two directions. First, it shows that in the realizable setting considered by Golowich and Livni, Littlestone classes can be learned with mistake bound $O(\log T)$ against an adaptive adversary. Second, in the agnostic setting with an adaptive adversary, such classes can be learned with regret $O(\sqrt{T})$.

The paper's realizable learner builds on the strategy of Golowich and Livni, but introduces two main refinements to address how their probabilistic analysis breaks in the presence of an adaptive adversary. The first idea is to apply lazy updates to ensure independence between examples revealed and the randomness used in their permutation argument. The second is to use uniform convergence to ensure accurate estimation for all functions in the target class, rather than just the ground truth labeling function.

In the agnostic setting, it provides two different algorithms that reduce private online learning to sanitization (a.k.a. private synthetic data generation). The first is a simple proper learner (albeit with suboptimal regret) that breaks the time horizon into batches, applies a sanitizer to each batch to produce synthetic examples, and runs a non-private online learner on the synthetic data. The second is an improper learner with improved regret that simulates the experts algorithm, evaluating experts using sanitization. These online learners can then be instantiated using the known private sanitizer of Ghazi et al. for Littlestone classes.

**Questions:**

* How close to proper can the algorithm handling the realizable case be made? E.g., by replacing SOA hypotheses with Hanneke-Livni-Moran's construction?

* I understand you're space-constrained, but seeing as Section 4.2 reflects most of the paper's algorithmic novelty, I would have liked to have seen a more detailed description of that algorithm in the main body of the paper in place of the algorithm described in Section 3.

**Ethical Concerns:**

["NO or VERY MINOR ethics concerns only"]

**Final Justification:**

This is a solid submission and the authors convincingly addressed questions and comments raised in the initial reviews.

**Limitations:**

Yes.

**Paper Formatting Concerns:**

None.

**Quality:**

4

**Strengths And Weaknesses:**

(+) Resolves the main open questions left by Golowich and Livni. Note that there was no reason to believe a priori that the best achievable mistake bound/regret of DP online learning against an adaptive adversary should coincide with that of an oblivious adversary; indeed, there are related settings in which there is a separation.

(+) The algorithms and analyses in Section 4 introduce some nice conceptual ideas that could be useful more generally.

(-) Handling the realizable case requires overcoming some technical challenges from prior work, but is otherwise relatively incremental. This isn't necessarily a strength or a weakness, but the paper's technical contributions don't have that much to do with understanding the combinatorial structure of Littlestone classes, and more to do with developing and analyzing online learners.

---

> ### Author Rebuttal · Authors · 2025-07-31
>
> We thank the reviewer for the constructive comments and provide our detailed responses below.
>
> > How close to proper can the algorithm handling the realizable case be made? E.g., by replacing SOA hypotheses with Hanneke-Livni-Moran's construction?
>
> By replacing the SOA with Hanneke-Livni-Moran's construction, we can obtain a learner whose output is always a majority vote of $O(d_V^\star)$ hypotheses from the given class, where $d_V^\star$ is the dual VC dimension.
>
> More generally, the output space of our algorithm is exactly the output space of the deterministic online learner it invokes. Thus, how close to proper the algorithm can be made depends on how close to proper a deterministic online learner can be made. For classes that can be properly learned by a deterministic online learner (e.g., thresholds over a finite domain), the resulting private online learner is also proper. However, there are some classes that cannot be properly learned by a deterministic online learner (e.g., point functions). For these classes, our algorithm (Algorithm 2) is improper.
>
> Additionally, our first algorithm for the agnostic setting (Algorithm 3) can also yield a proper learner. Though it works for all Littlestone classes, it incurs a large regret ($T^{3/4}$). We believe this can be improved and leave it as future work.
>
> We will include the above discussion in the revision.
>
> > I understand you're space-constrained, but seeing as Section 4.2 reflects most of the paper's algorithmic novelty, I would have liked to have seen a more detailed description of that algorithm in the main body of the paper in place of the algorithm described in Section 3.
>
> Thank you for your suggestion. We will allocate more space for Section 4.2 and provide a detailed description of our agnostic learner in the revision to better highlight our algorithmic contributions.

---

> > ### Comment · Reviewer_2ZzJ · 2025-08-06
> >
> > Thank you for addressing my comments. I have no further questions and maintain my positive impression of the submission.

---

### Official Review · Reviewer_NrgB · 2025-07-10

**Clarity:** 2
**Significance:** 3
**Originality:** 2
**Rating:** 5
**Confidence:** 3

**Summary:**

This work studies mistake/regret bounds for private online learning against adaptive adversaries. They improve over existing $O(\sqrt{T})$ mistake bounds for private realizable online learning of finite Littlestone classes against adaptive adversaries and obtain $\tilde{O}(\sqrt{T})$ regret bounds for private agnostic online learning of finite Littlestone classes against adaptive adversaries.

**Questions:**

See notes above.

**Ethical Concerns:**

["NO or VERY MINOR ethics concerns only"]

**Final Justification:**

I have a positive appraisal of the paper and the authors addressed my concerns with presentation by committing to update the paper to improve clarity.

**Limitations:**

Yes

**Paper Formatting Concerns:**

No concerns.

**Quality:**

3

**Strengths And Weaknesses:**

This paper makes a significant contribution to understanding mistake/regret bounds for private online learning against adaptive adversaries. While the main algorithmic idea is structurally similar to that of Golowich and Livni, there are notable technical obstacles to improving their mistake bounds against adaptive adversaries, which this work overcomes.

I unfortunately found this paper hard to read for a few reasons. To the author’s credit, they give a summary of of the online learning of Golowich and Livni and explain why it is insufficient for achieving the desired $O(\log T)$ mistake bounds, but I don’t know how helpful this summary would be for those who are not already familiar with the algorithm. More importantly, the reader has to do some digging to find a formal theorem statement for the algorithm in Section 3, and then more digging to find the setting of parameter $N_0$, which I think is helpful for the reader to know while reading the pseudocode for Algorithm 2. It would be good to have a theorem statement in Section 3 with the parameters necessary for privacy and the stated mistake bounds specified.

Some minor notes/typos:

In abstract - “each time step a hypothesis”

Theorem 2.6 - “pirvate histogram”

“incurring a mistake bound scales”

Algorithm 2 - “failuare”

I think there’s a duplication issue with the citation for GGKM “Sample Efficient Proper PAC Learning with Approximate Differential Privacy”

---

> ### Author Rebuttal · Authors · 2025-07-31
>
> We thank the reviewer for the constructive comments and provide our detailed responses below.
>
> > they give a summary of the online learning of Golowich and Livni and explain why it is insufficient for achieving the desired $O(\log T)$ mistake bounds, but I don’t know how helpful this summary would be for those who are not already familiar with the algorithm.
>
> We include a description of Golowich and Livni's algorithm because our work builds upon their framework (specifically, constructing tournament examples). A detailed explanation will help clarify how and why our algorithm works against an adaptive adversary. We will revise the presentation to enhance accessibility for non-experts.
>
> > It would be good to have a theorem statement in Section 3 with the parameters necessary for privacy and the stated mistake bounds specified.
>
> Thank you for your suggestion. We will include a formal theorem stating the privacy/utility guarantee of our algorithm and the choice of parameter $N_0$.
>
> > Some minor notes/typos:
>
> Thank you for identifying the typos. We will correct them in the revision.

---

> > ### Comment · Reviewer_NrgB · 2025-08-06
> >
> > Thank you for the response. I will maintain my score.

---

### Decision · Program_Chairs · 2025-09-17

**Decision:**

Accept (poster)

**Comment:**

While there exist algorithms that achieve a mistake bound of $O(\sqrt{T})$ against an adaptive adversary in the realizable setting for private online learning of Littlestone classes over a time horizon $T$, there also exist algorithms that achieve a mistake bound of $O(\log T)$ against an oblivious adversary. This paper closes the gap by giving an algorithm that achieves a mistake bound of $O(\log T)$ against adaptive adversaries. All reviewers agree that this is an interesting result since it was not clear whether there exists a separation between the models.

I would encourage the authors to address reviewer concerns and make improvements to the overall presentation, to strengthen the potential impact of the work